# The Clinical Pharmacology and Therapeutic Evaluation of Non-Steroidal Anti-Inflammatory Drugs in Adult Horses

**DOI:** 10.3390/ani13101597

**Published:** 2023-05-10

**Authors:** Melissa A. Mercer, Jennifer L. Davis, Harold C. McKenzie

**Affiliations:** 1Department of Biological Sciences and Pathobiology, Virginia Maryland College of Veterinary Medicine, Blacksburg, VA 24061, USA; 2Department of Large Animal Clinical Sciences, Virginia Maryland College of Veterinary Medicine, Blacksburg, VA 24061, USA

**Keywords:** NSAIDs, horses, clinical pharmacology, endotoxemia, osteoarthritis

## Abstract

**Simple Summary:**

Non-steroidal anti-inflammatory drugs (NSAIDs) are commonly prescribed in equine practice for a variety of painful or inflammatory conditions. This review discusses the clinical pharmacology of NSAIDs in adult horses and methods of evaluating the therapeutic effect of these drugs in equine research.

**Abstract:**

This review firstly examines the underlying pathophysiology of pain and inflammation associated with orthopedic disease and endotoxemia. Then, it reviews the clinical pharmacology (pharmacokinetics and pharmacodynamics) of both conventional and non-conventional NSAIDs in the adult horse, and finally provides an overview of different modalities to evaluate the therapeutic efficacy of NSAIDs in research.

## 1. Introduction

Non-steroidal anti-inflammatory drugs (NSAIDs) are commonly prescribed in equine practice for a variety of painful or inflammatory conditions. The prescribing activity of NSAIDs varies worldwide, with one study examining electronic record systems reporting NSAID prescriptions in 42.4% of cases in the US, with lower prescribing rates in Canada (34.2%) and the UK (28.6%) [1]. Studies evaluating NSAIDs in horses typically fall into three main categories: pharmacokinetic studies, pharmacodynamic/efficacy studies, and safety studies. Pharmacokinetics uses mathematical equations to describe the activity and time course of a drug in the body and includes evaluation of the absorption, distribution, metabolism, and elimination of the drug [2]. Pharmacodynamics is the evaluation of a drug’s molecular, physiologic, and biochemical effects or actions in the body [2]. Together, pharmacokinetics and pharmacodynamics explain the dose–response relationship that is observed clinically. To properly evaluate therapeutics, the researcher must have knowledge of the physiology of the disease that treatment is targeting, the clinical pharmacology of the therapeutic examined, and the pros and cons of the different modalities used to evaluate therapeutic efficacy. This review will examine the underlying pathophysiology of pain and inflammation associated with two of the most common conditions for NSAID prescription—orthopedic disease and endotoxemia—and will provide a review of the clinical pharmacology of both conventional (i.e., non-selective and COX-2-selective) and non-conventional NSAIDs in the adult horse, as well as an overview of different modalities to evaluate the therapeutic efficacy of NSAIDs in research.

## 2. Pain, Nociception, and the Cyclooxygenase Pathway

### 2.1. Nociception

Horse owners are very confident in their own ability (87.7% of respondents) and their veterinarian’s ability (91.2%) to accurately recognize pain in horses [3]. The perceived ability to correctly recognize pain in their horses leads to many horse owners obtaining analgesic drugs from both veterinary–patient–client relationship (VPCR) compliant sources and non-VPCR compliant sources [4]. For a veterinarian to best understand and treat pain, it is critical to know the underlying cause and physiology of pain. It is important to recognize that pain is a combined sensory and emotional experience that is separate from the activity of sensory neurons (i.e., nociception) [5]. Nociception encompasses the action of transduction, transmission, modulation, and central processing of potential, or actual, noxious or painful stimuli [6]. The central processing of these stimuli ultimately results in the subjective sensory experience that is pain.

Inflammatory pain, for example, is evoked by a variety of stimuli secondary to tissue injury and is commonly associated with wounds, musculoskeletal injuries, and intestinal inflammation in horses. Inflammatory pain serves to promote healing and protect the body during the healing process by producing a hypersensitivity state that parallels the duration of active inflammation [7]. In inflammatory pain states, the presence of a noxious stimulus (thermal, mechanical, chemical, etc.) leads to the activation of nociceptive neurons that release action potentials at high-threshold nociceptors, which then convert (transduce) the stimuli to electrical impulses that are transmitted by thinly myelinated (Aδ) and unmyelinated (C) primary sensory nerve fibers [8]. The thin myelination of Aδ nerve fibers allows for the quick transmission of impulses, producing the sharp, acute pain that stimulates rapid withdrawal from a stimulus [9]. In contrast, the unmyelinated C-fibers have slower transmission of electrical impulses. In horses, action potentials associated with visceral pain are transmitted by C fibers within the autonomic nervous system [10].

The action potential is transmitted to the dorsal horn of the spinal cord, where signals are further carried to the brain by neurons that travel ascending pathways. The spinothalamic tract carries action potentials to the thalamus, where discernment of the intensity and location of the pain source occurs [11]. The spinoreticulothalamic tract, in contrast, carries action potentials to the brainstem, which is associated with poorly localized pain [8,12]. Finally, these signals are transmitted to higher cortical structures, such as the somatosensory cortex and the anterior cingulate gyrus, that aid in the processing of sensory and emotive experiences or pain, respectively [13,14].

The central sensitization associated with chronic pain states is caused by the production and distribution of inflammatory tissue byproducts, causing repetitive stimulation of the peripheral sensory nerves resulting in the summation and production of prolonged action potentials in dorsal horn sensory neurons of the spinal cord [9]. Some examples of inflammatory tissue byproducts include prostanoids, peptides (bradykinin, substance P, and CGRP), cytokines (IL1β and IL6), eicosanoids, endocannabinoids, leukotrienes, and ions (hydrogen and potassium) [8,9]. Central sensitization is what allows low-intensity stimuli to produce pain, contributing to changes in sensory processing in the spinal cord that lead to the release of inhibitory neurotransmitters from descending pathways of the spinal cord, such as 5-hydroxytryptamine (serotonin), norepinephrine, and endogenous opioids [9]. Central sensitization is further exacerbated in nerve injury by activation of the microglia and astrocytes that function to release cytokines (such as TNF-α, IL1β, and IL6), which serve to further increase central sensitivity to pain [8].

### 2.2. Introduction to Prostanoids and Cyclooxygenases

Prostanoids are cyclooxygenase-mediated biosynthesized metabolites of arachidonic acid that act with paracrine or autocrine functions in a variety of health and disease states and are some of the primary mediators of inflammatory pain [15,16]. The most well-studied and principal pro-inflammatory prostanoid generated from the arachidonic acid cascade is PGE_2_, which interacts with both peripheral and central nociceptive pathways [17]. Other local activities of PGE_2_ include increasing sensitivity to bradykinin or capsaicin on peripheral nerves, resulting in lasting hyperalgesia to mechanical and chemical stimuli [17,18]. Centrally, PGE_2_ increases dramatically within the spinal cord following inflammation or nociception, where it facilitates the release of glutamate, substance P, and CGRP [19,20]. Other prostanoids implicated in inflammatory pain include PGI_2_ and PGD_2_. PGI_2_ production has also been found to rapidly increase at the site of tissue injury following inflammatory stimuli, which is chiefly modulated through the PGI_2_-IP pathway [17]. PGD_2_, however, has demonstrated both pro- and anti-inflammatory properties in tissues [17]. PGD_2_ has been implicated in promoting PGE_2_-mediated pain states, as PGDS-knockout mice did not exhibit allodynia associated with intrathecal PGE_2_ administration but continued to exhibit thermal sensitivity [21].

COX enzymes are membrane-bound proteins that are produced by the endoplasmic reticulum and catalyze bis-oxygenase and peroxidase reactions in the arachidonic acid cascade [22]. The bis-oxygenase reaction of COX enzymes functions to convert arachidonic acid to Prostaglandin (PG) G_2_, while the peroxidase reaction coverts PGG_2_ to PGH_2_. PGH_2_ then undergoes isomerization, oxidation, or reduction by an assortment of prostanoid synthases to generate PGE_2_, PGI_2_, PGD_2_, PGF_2α_, and Thromboxane A_2_ (Figure 1).

The primary COX enzymes present in mammalian species include COX-1 and COX-2. COX-1 is expressed in most tissues, and is widely considered to be the constitutive isoform, maintaining a variety of housekeeping functions, including renal blood flow, gastrointestinal mucosal integrity through production of PGE_2_, and platelet aggregation through production of thromboxane A_2_ [23]. While COX-2 has been historically described as an enzyme that is induced in proinflammatory states in most tissues, this statement has been refuted in recent years, as research has uncovered COX-2’s homeostatic activities in the gastrointestinal, renal, and cardiovascular systems [17].

Recent research performed in COX-1-deficient mice has begun to elucidate the complex interactions between COX-1 and inflammatory states. While COX-2 is the predominant enzyme in the induction of a prostaglandin response to lipopolysaccharide (LPS), increased COX-1 expression was also found in the circulating monocytes of mice treated with LPS, demonstrating induction of COX-1 expression as a direct result of a profound inflammatory insult [24]. COX-1 induction has also been implicated in the modulation of neuropathic and post-operative pain, where intrathecal administration of a non-selective COX inhibitor resulted in a delay in the onset of hypersensitivity following peripheral nerve injury in rats and intrathecal administration of a COX-1 selective inhibitor resulted in complete blockade of the development of hypersensitivity [25]. Furthermore, additional studies of peripheral nerve injury in rats demonstrated a time- and location-dependent change in expression of COX-1 in the spinal cord in response to injury, suggesting that COX-1 plays a role in modulating the central sensitization and processing of pain [26].

In contrast to COX-1, the role of COX-2 enzyme induction in inflammatory states is much better recognized. COX-2 induction primarily begins at the local tissue level with inflammation, where high levels of COX-2 have been found in injured cells, as well as macrophages, neutrophils, and mast cells which have infiltrated the site of tissue injury in murine models [17,27]. COX-2 also has central activity with pain states; induction of expression of COX-2 in the dorsal root ganglion has been found with periganglionic or peripheral nerve inflammation, where it was not present prior to inflammatory stimulation [17,28]. The central activity of COX-2 may also be mediated through its interaction with the endocannabinoid system. COX enzymes have been shown to metabolize the endocannabinoids AEA and 2-AG, which contributes to the short half-life of these compounds in inflammatory states [29]. Therefore, it has been postulated that COX inhibition could promote a longer half-life of endocannabinoids centrally by decreasing endocannabinoid breakdown [30]. While the induction of COX-2 in inflammatory pain states has been well established, the underlying downstream mechanistic effects following this induction remain incompletely elucidated.

## 3. Disease Processes Targeted by NSAIDs

### 3.1. Endotoxemia and Pyrexia

The systemic inflammatory response associated with endotoxemia remains a leading cause of morbidity and mortality in both neonatal and adult equids. While endotoxemia specifically refers to circulating endotoxin (a cell wall component of Gram-negative bacteria), the term has commonly been used more broadly as a clinical description of horses experiencing systemic inflammatory response syndrome (SIRS) secondary to sepsis, described below [31].

Endotoxin is a lipopolysaccharide (LPS) consisting of three main structural components: the hydrophobic lipid A, a core oligosaccharide, and the hydrophilic antigen O. Large numbers of Gram-negative bacteria are present in the equine intestinal tract, particularly the colon, and are prevented from reaching systemic circulation in significant quantities by intestinal epithelial tight junctions. The small number of Gram-negative bacteria that do enter the systemic circulation from the equine GI tract travel through the portal vein to the liver where they are neutralized by a combination of circulating anti-LPS antibodies and hepatic mononuclear phagocytes. However, intestinal inflammation or extra-intestinal infections can lead to significant quantities of Gram-negative bacteria and/or LPS entering systemic circulation through either overwhelming or bypassing hepatic clearance mechanisms.

When LPS is released into circulation following the rapid multiplication or death of Gram-negative bacteria, it is transported by hepatically synthesized LPS binding protein to the pattern recognition receptor CD14, where it interacts with toll-like receptor 4 (TLR4) [32]. Interaction between LPS, LPS binding protein, CD14, and TLR4 leads to NF-kB-mediated downstream upregulation of pro-inflammatory mediators, such as TNF-α, eicosanoids, interleukins, and tissue factors [31]. This rapid influx of pro-inflammatory mediators and the pro-inflammatory immune response define the clinical systemic inflammatory response syndrome (SIRS) and they are initially advantageous in clearing localized infections.

The most prominent pro-inflammatory cytokines in equine SIRS are TNF-α, IL-1β, and IL-6, all of which are also regulators of innate immunity [33]. An increase in TNF-α has been positively correlated with the clinical signs of SIRS in horses, including an increase in body temperature, increase in heart rate, and decrease in total white blood cell (WBC) count, as well as an increase in mortality rate in horses with strangulating or inflammatory intestinal disorders [34,35]. TNF-α has been known to induce IL-1β production and release by macrophages [36], and IL-1 subsequently induces production of IL-6 in a paracrine fashion [37]. In horses, exposure to LPS increases IL-1β gene expression in peripheral leukocytes [38], and expression of IL-1β is positively correlated with an increase in rectal temperature and increased expression of other cytokines (IL-8 and IL-10) [33]. Following LPS infusion to horses, there was also an increase in circulating IL-6 [33], and an increase in systemic and peritoneal IL-6 has been associated with strangulating or inflammatory intestinal disorders, endotoxemia, and non-survival [39]. Since prostaglandins have been shown to affect the formation of pro-inflammatory cytokines, such as IL-6 [40], and IL-1β has been shown to induce mRNA expression of COX-2 [41], COX-inhibitors remain central to the treatment of the profound systemic inflammatory response associated with endotoxemia.

LPS-induced pyrexia is typically a tri-phasic response, with an initial febrile response elicited within 30 min of exposure, followed by secondary and tertiary febrile responses 1.5–12 h later. The initial febrile response is mediated by hematopoietic cell TLR4 recognition of LPS, resulting in the upregulation of pulmonary and hepatic PGE-2-synthesizing enzymes [42]. The secondary and tertiary phases of the LPS fever response are governed by upregulation of microsomal PGE synthase-1 and COX-2 in the brain, resulting in central PGE_2_ production [43]. Therefore, COX-produced prostaglandins are also an important therapeutic target in the management of the clinical signs associated with endotoxemia in horses.

### 3.2. Osteoarthritis

Lameness is one of the most common complaints veterinarians attend to in horses, and osteoarthritis (OA) is the leading cause of lameness in horses worldwide, accounting for approximately 60% of lameness cases in adult horses [44]. NSAIDs remain a cornerstone of treatment of chronic orthopedic conditions in the horse due to their availability, lack of potential for human abuse, and low cost. Osteoarthritis diagnosis in horses leads to increased risk of early retirement from competition, delayed return to normal activity, and increased costs and morbidity associated with treatment [44]. Osteoarthritis is considered a structural disease of the entire joint, resulting in progressive, permanent deterioration of multiple joint tissues, such as cartilage, subchondral bone, and synovium [45]. Multiple factors can lead to its development in the horse, including trauma, conformation, age, and husbandry practices (i.e., shoeing) [46]. While there are several different pathways for OA development in horses, the clinical presentation of OA is consistent despite wide inter-patient variability in joint pathology.

Disease-modifying drugs are agents which have been proven to halt the progression or reverse the pathology associated with disease [47]. There are no FDA-approved disease-modifying drugs for the treatment of OA in horses, and therefore treatment is limited to managing the pain and inflammation associated with OA through the administration of symptom-modifying drugs [48]. In horses, the primary presenting clinical signs of OA are lameness resulting from pain and limited function associated with the affected joint [49]. Initial pain in arthritic animals occurs primarily during loading of the affected joint and is characteristic of nociceptive pain. Joints are innervated by a combination of high-threshold Aδ and C-fibers that typically only respond to strong rotational or impact pressure. However, joint disease, such as OA, results in mechanical hyperalgesia or the lowering of the excitation threshold for nerve fibers, resulting in enhanced pain sensitivity of the affected joint [50].

As disease progresses, OA pain is characterized as a chronic pain state that is dependent on these combined peripheral and central nociceptive pathways, which may result in neuropathic and maladaptive pain states in certain individuals. The presence of a neuropathic pain state is important, as this suggests direct damage to neurons innervating the joint, leading to a pain state that is independent of the severity of underlying joint pathology [50]. Therefore, modification of peripheral and central nociceptive pathways is essential for the adequate management of chronic equine OA. NSAID inhibition of cyclooxygenase (COX)-produced prostaglandins represents an important therapeutic target in the management of clinical signs, as well as reduction in the production and propagation of the inflammatory response associated with osteoarthritis and other orthopedic disorders in horses.

## 4. Experimental Models Used to Evaluate Pain and Inflammation in Equine Research

### 4.1. Evaluation of Equine Pain

Evaluation of pain behaviors is an inherently subjective task, as pain is a complex, multi-dimensional emotional experience encompassing psychosomatic and physiologic alterations in response to noxious stimuli. Horses in pain exhibit several altered behaviors, including changes in weightbearing, head lowering, anti-social/depressive behaviors, self-mutilation, rolling, pawing, poor performance, and changes in facial expression [51]. While both owners and veterinarians are undoubtedly utilizing these behaviors to determine if their animals are in pain and if they require therapeutic intervention, the ability to quantify a change in pain behavior to evaluate the efficacy of therapeutic intervention in prospective studies remains a challenge. In an effort to create quantifiable, unbiased, and reproducible methods of evaluating equine pain behavior, a small number of behavior-based pain scales have been developed and validated for use in horses. Three scales have been developed for the evaluation of equine facial expressions—the Equine Pain Face Scale [52], the Equine Utrecht University Scale of Facial Assessment of Pain [53] for somatic pain, and the Horse Grimace Scale [54] for visceral pain. These rubrics include evaluation of ear position, as well as orbital, nostril, muzzle, and facial mimic muscle tension. However, the repeatability and reproducibility of such behavioral scales has been questioned, as behavioral expressions of pain can vary based on personality and environmental stimuli, and in general facial expressions were found to be less reliable than body behaviors in horses with induced orthopedic pain [55]. Additionally, since most pain scales have been developed for animals experiencing moderate–severe acute pain, the reliability and inter-rater agreement may be less robust in mild acute pain states or in animals experiencing chronic, low-level pain [55].

A composite pain scale (CPS) has been developed in an effort to combine both behavioral assessment (interaction, response to stimuli, posture, and reaction to palpation) and physiologic parameters (heart rate, respiratory rate, and rectal temperature) to better bridge the gap between subjective and objective evaluation [56]. The CPS has acceptable inter- and intra-rater agreement (k > 0.8) [55,56]; however, in examining horses with induced lameness, temperature, heart rate, and posture were the factors retained in the model best describing movement asymmetry [55]. Given that the major factors that best describe movement asymmetry were physiologic parameters and that the CPS requires 5 min of live observation per data collection point, the utility of implementing this scale in combined pharmacokinetic/pharmacodynamic or efficacy studies with rapid sampling windows is questionable. Therefore, due to these challenges, most studies forgo generalized pain behavior evaluation and opt to use the physiologic parameters and disease-specific scales described above to evaluate equine pain and its response to treatment.

### 4.2. Models Used in Equine NSAID Pharmacodynamics and Efficacy Studies

#### 4.2.1. Mechanical Nociception Models

Mechanical nociception is induced through the application of force over a selected area. One of the most-validated methods of measuring mechanical nociception is through either hand-held or wireless pressure algometers. Algometers use a blunt-ended pin and a pneumatic piston to sequentially apply pressure over a selected area (usually the dorsal metacarpus or withers of the horse) until withdrawal of the limb or behavioral response occurs [57]. The advantage of these models is that they are standardized, repeatable, and provide immediate, objective, quantitative data for evaluation. However, these models present challenges with respect to their use in clinical studies, as the equipment is specialized, expensive, does not directly evaluate the clinical outcome measure of lameness, and some systems may allow for individual variation in terms of their application. Additionally, the behavioral effects of some medications (such as increased spontaneous motor activity with opioids) may preclude accurate evaluation of the nociceptive response with these systems [57].

One of the most popular mechanically induced lameness models in equine pharmacology is a reversible sole-pressure model, typically utilizing adjustable heart-bar or set-screw shoes [58,59,60,61,62]. The challenge with the reversible sole-pressure model in evaluating NSAIDs is that this model induces pain states through activation of cutaneous nociceptive fibers in the hoof sole, as there was no evidence of systemic inflammation following lameness induction [63]. However, it should be noted that since this study only examined systemic markers of inflammation, a localized inflammatory response to this model could not be ruled out. Based on the findings of this study, the utility of sole-pressure models in evaluating drugs that provide analgesia primarily through anti-inflammatory pathways has been questioned.

Despite the potential pitfalls, sole-pressure models have been used successfully in evaluating the efficacy of both non-selective and COX-2 selective inhibitors, both of which primarily exert their analgesic effects through anti-inflammatory activity [58,59,60,61,62]. Phenylbutazone was more effective in reducing lameness when administered orally (4.4 mg/kg) in horses in the adjustable heart-bar shoe model compared to placebo and meloxicam, with a maximal clinical response at least 4 h post-treatment [64]. Similarly, flunixin meglumine was effective in reducing lameness for up to 12 h post-IV administration (1.1–2.2 mg/kg) when compared to a half-label dose of flunixin meglumine (0.5 mg/kg) [65]. Sole-pressure models have also been used to compare the efficacy of non-conventional NSAIDs, such as acetaminophen, to conventional COX inhibitors in horses [60,61,62]. The advantages of sole-pressure models in horses are that they are cost-effective, reversible methods of inducing lameness that do not require specialized equipment and do not result in lasting injury or impairment of future performance.

#### 4.2.2. Adjuvant and Injury-Induced Lameness Models

Adjuvant and injury models are commonly used to mimic naturally occurring synovitis and osteoarthritis for the experimental induction of lameness in equine research.

To evaluate the efficacy of therapeutics for the treatment of synovitis and synovial sepsis, the most-used model is the intra-articular LPS-induced synovitis model. In this model, LPS (0.125–300 ng) is injected into joints, typically metacarpophalangeal, radiocarpal, or intercarpal joints [64,66]. This model is advantageous in that it reliably produces lameness and joint inflammation (effusion and heat) within 2 h of LPS administration, which subsequently resolves by 24–48 h post-administration [66]. This model has typically been used to examine the effects of NSAIDs on intrasynovial inflammatory mediators and cartilage biomarkers [67,68]. However, this model has also been used to examine the effects of oral phenylbutazone and meloxicam on lameness using a body-mounted inertial sensor system [64]. Meloxicam significantly reduced the total difference in head height in comparison to placebo at 6 and 8 h on objective gait analysis; this was indicative of significantly less variation between the right and left portions of the stride of the forelimb gait in meloxicam-treated horses compared to placebo [64]. There were no significant changes in lameness for the total difference in head height for phenylbutazone-treated horses in comparison to the meloxicam treatment or placebo [64].

One of the most common models used to mimic naturally occurring osteoarthritis in the horse is the osteochondral fragment model. In this model, an osteochondral fragment is traumatically created through arthroscopic surgery, typically of the metacarpophalangeal, metatarsophalangeal, or carpal joint. The primary advantage of this model is that it is the one that most closely reflects natural osteoarthritis development and pain in the horse [69,70]. This model may be performed in multiple joints simultaneously, which can be advantageous when evaluating the efficacy of local joint therapy, as each horse can serve as its own internal control, leading to reduction in the number of horses required [69]. However, this model would not lead to subject-number reduction for the administration of systemic therapies, and while lameness can be reversible depending on the severity of the lesion induced and the removal of the created fragment, it is likely that osteoarthritis will progress over time in the affected joints [69,70]. Additionally, this model requires 2 weeks from model induction until lameness becomes apparent on straight-line kinetic gait analysis [69] and BMIS [71] or on subjective evaluation following joint flexion [69], and 20 days for lameness to be apparent on subjective lameness evaluation when circling [71]. It takes up to 70 days following model induction for lameness to be evident with subjective lameness evaluation in a straight-line trot without joint flexion [71]. Therefore, due to the protracted study duration required, the expense of the surgical procedures, and the morbidity and mortality associated with these procedures, this model is limited to pilot-study application. In addition to these models, there is also a surgical tendinitis model and a variety of other models that have been described in the horse [72,73]. However, these models are beyond the scope of this review, as they are primarily used for investigating the utility of biologics in the treatment of soft-tissue injuries in the horse.

#### 4.2.3. Models of the Equine Systemic Inflammatory Response

To evaluate the efficacy of therapeutics in the treatment of laminitis resulting from a profound systemic inflammatory response specifically, the three major models used are the black walnut extract (BWE) model, the carbohydrate overload (CHO) model, and the euglycemic-hyperinsulinemic clamp (EHC) model. The BWE model involves nasogastric administration of soaked black walnut hardwood shavings (2 g/kg), resulting in a systemic increase in pro-inflammatory cytokines within 1.5 h, profound neutropenia within 4 h, and decreased laminar blood flow and increased lameness within 12 h of administration [74]. In this model, most horses recover within 6 days of model induction; however, some horses have developed significant morbidity and mortality [74]. The CHO model involves nasogastric administration of the soluble carbohydrate oligofructose (10 g/kg), resulting in increase in heart rate within 4 h, increase in rectal temperature within 8 h, increase in lameness within 24–36 h, and neutrophilia and leukocytosis within 20–48 h [75]. In the EHC model, hyperinsulinemia (>1000 µIU/mL) is achieved while maintaining euglycemia (5 mmol/L) using a modified euglycemic-hyperinsulinemic clamp technique for up to 72 h, resulting in clinical signs of laminitis within 72 h of model induction [76]. When performing studies to evaluate treatment efficacy, model selection is critical; the BWE and CHO models appear to better mimic laminitis associated with severe endotoxemia/SIRS [77], while the EHC model best mimics endocrinopathic laminitis [76].

In addition to the SIRS/laminitis models above, the most common model for evaluation of the response of systemic inflammation to drug treatment in horses is the experimentally induced endotoxemia model. In this model, LPS (*E. coli* 055:B5) is administered intravenously (30–500 ng/kg), typically as a CRI over 0.5–6 h. Following LPS model induction, peak serum TNF-α concentrations are achieved within 1.5 h [34], neutropenia and leukopenia within 2–4 h [78], and peak fever response within 2–4 h and lasting up to 12 h [79]. The LPS model is reversible and at lower doses is not associated with long-term adverse effects. However, a challenge with this model is that there is a risk of the development of tolerance to LPS, particularly in crossover studies with short washout periods between crossovers. Tolerance manifests as lower peak rectal temperature, shorter duration of temperature elevation, decrease in the magnitude and duration of TNF-α elevation, lower heart rate, and decreased pain scores compared to naïve LPS exposure [79,80]. The duration of in vivo endotoxin tolerance has been examined in horses, and tolerance lasts a minimum of 7 days but as long as 21 days post-exposure [80]. Therefore, appropriate washout periods and statistical analyses must be accounted for when performing cross-over studies using an LPS model.

## 5. Experimental Methods Evaluating the Therapeutic Efficacy of NSAIDs in Horses

### 5.1. Subjective Lameness Evaluation

The oldest method of lameness determination is subjective (visual) lameness evaluation. However, the challenge with subjective lameness evaluation is to create a method of evaluation that is defined, repeatable, and reproducible between observers with acceptable inter- and intra-observer agreement. Furthermore, the ideal scale must consider the variety of circumstances in which lameness may be observed, whether it is more apparent with certain gaits, under certain surface conditions, or under certain movement patterns (straight line, circling, under saddle, etc.). Unfortunately, the use of these scales in clinical research is mired in complications, and the statistical analysis performed often does not account for ordinal data through the use of non-parametric statistical tests [81].

Therefore, in an attempt to standardize subjective lameness evaluation, a number of different lameness scoring rubrics have been developed with varying degrees of adoption amongst practitioners. The AAEP scale is graded 0–5, with clear and unambiguous guidelines in both the classification of lameness and the conditions under which lameness should be evaluated [82]. The challenge with this scale, however, is that consistent lameness at a trot that is not obviously apparent at a walk will always be classified as grade 3 lameness, regardless of the severity of lameness observed. Therefore, the AAEP scale is relatively insensitive in determining finer changes in severity in response to treatment or diagnostic analgesia [83]. This has led to challenges in agreement between independent observers utilizing this scale. Agreement between observers for identifying lameness using the AAEP scale on a straight-line trot was fair to moderate (k = 0.44), which only slightly improved following complete lameness evaluation (k = 0.45) [84]. When localizing lamenesses using the AAEP scale, agreement was slightly higher for forelimb lamenesses (k = 0.52) than hindlimb lamenesses (k = 0.38), and agreement was better for more severe, consistent lameness (AAEP score > 1.5, k = 0.86) than for mild, inconsistent lameness (AAEP score < 1.5, k = 0.23) [84].

In response to these challenges in delineating lameness severity amongst observers, a 10-point scale has been used—particularly in countries outside the US. The 10-point scale is graded 0–9 and has similar unambiguous guidelines for the evaluation of lameness [85]. The ability to delineate lameness severity more finely with this scale was shown in the agreement between three experienced observers using the 10-point scale (k = 0.41) for mild lameness (grades 0–4) being higher than that for similar grades (<1.5/5) using the AAEP scale (k = 0.23); however, agreement declines amongst inexperienced observers [84,86]. A global scoring system has also been developed to evaluate change in lameness from the baseline in response to repeated evaluation, diagnostic analgesia, or therapeutic intervention [86]. For the global scoring system for change in lameness as a result of repeated lameness evaluation, the agreement between reviewers (k = 0.6) was better than that for the 10-point lameness scale (k = 0.41) [86]. In addition, there are a number of other scales that are less frequently used and often practitioner-specific, which increases the challenge in describing lameness between practitioners, particularly if a horse enters another veterinarian’s care [83]. Therefore, while subjective lameness scoring is the most accessible and most commonly used tool for assessing lameness in the horse, it is a challenge to extrapolate findings between practitioners and to objectively quantify changes in lameness in response to treatment.

### 5.2. Objective Lameness Evaluation

#### 5.2.1. Body-Mounted Inertial Sensor Systems

Body-mounted inertial sensor (BMIS) systems are the most popular objective lameness evaluation metrics used in clinical practice. BMIS systems use kinematic techniques to evaluate lameness through determination of the effect of ground reaction forces on the stance phase of the stride [87]. The advantage of quantitative lameness evaluation is in the elimination of bias and its high degree of reproducibility, particularly in instances of repeated lameness evaluation.

BMIS systems analyze head and pelvic acceleration signals that are then converted into position trajectories following removal of background noise or extraneous movement [88]. The differences between relative maximal and minimal head and pelvic vertical height with respect to the right and left portions of the stride are then calculated. The software then determines the severity (based on amplitude of lameness), the laterality of the lameness, the stride phase associated with lameness (based on stance duration), and confidence in the evidence of lameness (based on stride–stride variation) [88]. The greatest clinical utility of BMIS systems appears to be in the initial diagnostic phase of lameness evaluation (i.e., limb localization), evaluation of the response to flexions, and the response to diagnostic analgesia testing. The advantage of using BMIS systems in efficacy and pharmacodynamic studies is that they provide an objective and reproducible standardized method of lameness evaluation. BMIS systems ultimately provide robust quantitative data that can be used in pharmacodynamic calculations.

The challenge with BMIS systems, however, is that they are designed for straight-line evaluation, and therefore may not be of utility for lamenesses which appear under certain stress conditions (turns and slopes). In a large-scale analysis of 1224 horses evaluated using the most common BMIS system (Equinosis Q with Lameness Locator^®^ software, Equinosis LLC, Columbia, MO, USA), the most frequent BMIS-assessed lameness category was combined forelimb and hindlimb lameness, accounting for 56.6% of diagnoses [88]. However, following complete lameness evaluation, the combined forelimb and hindlimb lameness category was the least common definitive diagnosis (10.9%) of horses for which a definitive diagnosis was achieved [88]. This lack of agreement can likely be explained by the law of sides. The first principle of the law of sides is the ipsilateral principle, where horses with a primary hindlimb lameness will weight-shift onto the opposite forelimb while trotting to offload the affected hindlimb [89]. The second principle of the law of sides is the contralateral principle, where a horse with a primary forelimb lameness may reduce their push-off force in the contralateral hindlimb to reduce the impact load on the affected forelimb [89]. Therefore, based on these two principles, BMIS systems may mis-identify an impact-driven forelimb force pattern or decreased push-off force hindlimb pattern as a lameness rather than as a compensatory movement pattern. Both compensatory movement patterns will be reduced or eliminated with improvement or resolution of the primary limb lameness. Additionally, a small proportion of cases in this study (9.6%) had no lameness that was identified on BMIS evaluation at a straight-line trot [88]. However, for horses where no lameness could be identified via BMIS, a definitive diagnosis was achieved in 44.6% of those cases following other components of the clinical lameness evaluation [88].

In terms of their agreement with conventional subjective lameness scoring rubrics, BMIS systems have been shown to be more sensitive in detecting the onset of mild lameness compared to veterinary evaluation in an induced sole-pressure model [87]. In a model of induced carpal OA, BMIS systems and subjective lameness scoring rubrics were found to have better agreement in identifying the lame limb than subjective lameness scoring evaluation and force-plate analysis [90].

#### 5.2.2. Force-Plate Analysis

Force plates are one of the oldest systems used to quantify lameness and are considered a gold standard in research settings for kinetic lameness quantification in a variety of species [90]. In these systems, lameness is quantified by both the stance duration and the peak vertical force applied to the plate. However, due to the nature of the data-capture device, only one foot strike on one limb can be recorded at a time, and the cost and specialized facilities required to install the unit make it impractical for clinical application. In a study of horses with induced carpal OA that compared subjective lameness scoring evaluation with two objective lameness evaluation systems (force plates and BMIS systems), the BMIS system was most accurate at determining the lame limb (60%), followed by subjective lameness scoring (51%), and force plates were the least accurate (42%) [90]. Additionally, agreement in lameness was better between BMIS and subjective lameness evaluation than subjective lameness evaluation and force-plate analysis [90]. Due to these pitfalls, force plates have largely been supplanted by kinematic analysis, particularly BMIS systems, in both research and clinical settings.

### 5.3. Objective Quantification of Pain and Inflammation

#### 5.3.1. Heart Rate and Heart Rate Variability

Both heart rate and, to a lesser extent, heart rate variability have been examined as outcome measures in evaluating the efficacy of NSAIDs in equine orthopedic and infectious disease models. The correlation between heart rate and other behavioral and physical evaluations of pain in horses remains generally weaker than more robust measures of the autonomic response, such as heart rate variability (HRV) or cortisol and catecholamine quantification [91]. The reason for this is that heart rate is easily influenced by behavioral or environmental factors [92]. Therefore, other measures, such as HRV, may be more robust in evaluating the autonomic response in horses.

Heart rate variability analysis is performed by analyzing echocardiogram tracings to describe the variation in amplitude and duration between successive heart beats, allowing it to serve as a quantitative marker of autonomic nervous system activity. Frequency-spectrum-domain measures are used to describe parasympathetic (vagal; HF), sympathetic (LF), and sympatho-vagal balance (VLF, LF:HF) by determining the amount of signal that lies within each frequency band [93,94]. There is a tendency toward an increase in LF and a decrease in HF or an increased LF:HF ratio in pain or stressed states, indicating an altered sympatho-vagal balance. The VLF band also describes sympatho-vagal balance, has been associated with cardiovascular disease prognosis and chronic inflammation in humans, and can be used as a predictor of prognosis in human HRV analysis [95,96]. The frequency-domain measures have been correlated with changes in Obel score and weight shifting in horses with laminitis; in particular, a decrease in HF has been correlated with an increase in weight-shifting frequency and adrenalin and noradrenalin concentrations—indicative of decreased vagal tone as pain increases [93]. NSAID administration was found to increase HF (vagal tone) and decrease LF (sympathetic tone) and weight-shifting frequency in laminitic horses [93]. The advantage of utilizing HRV in pharmacodynamic and efficacy studies is that it is a non-invasive measure of stress and pain in horses that provides non-biased quantitative data for analysis. The utilization of HRV in pharmacodynamics and efficacy studies may be affected by pharmacologic blockade of the autonomic nervous system (ANS) by specific drug action. In these cases, it may be advantageous to use HRV as a measure of efficacy for drugs with known effects on the ANS, such as atropine and its time-dependent decrease in LF [97] or propranolol with its time-dependent increase in LF [98]. However, if the action of a particular drug on the ANS is unknown, or if the HRV of a drug that acts on the ANS is compared to placebo, then results may be skewed because of pharmacological activity rather than stress or pain. Other challenges associated with the use of HRV includes the expense of the systems, the training required for application and analysis of the data, the time-consuming analytical process, and the potential effects of age, breed, feed status, and circadian rhythm on HRV [99].

#### 5.3.2. Cytokines and Acute-Phase Proteins

To accurately characterize the inflammatory response and the efficacy of NSAID therapy in response to systemic and local inflammation, cytokines are frequently measured in NSAID pharmacodynamic and efficacy studies. The cytokines that elevate in response to systemic inflammation in the horse, described in more detail earlier in this review, are TNF-α, IL-1β, IL-6, and IL-8, while IL-10 is the principle anti-inflammatory cytokine. Following infusion of LPS in healthy horses, gene expression of IL-1β, IL-8, and TNF-α peaked within 60 min, while IL-6 peaked at 90 min post-infusion [38]. There are a variety of commercially available equine-specific assays that have been validated for the measurement of equine cytokines, including fluorescent microsphere immunoassay kits [100] and multiplex assays [101,102], as well as more advanced molecular diagnostics, such as PCR. The challenge with these assays is that they require training in appropriate sample collection, preparation, and analytical technique.

To specifically determine the pharmacodynamics of NSAIDs in respect to their mechanism of action, quantification of COX-1 and COX-2 inhibition is valuable—particularly when performed in vivo. COX-1 and COX-2 activity are quantified using their surrogate markers, thromboxane B2 and PGE_2_, respectively, with commercially available kits [103]. Determining COX-1 and COX-2 inhibition is helpful when evaluating NSAIDs, as the degree of COX-1: COX-2 selectivity can be determined by the IC50 or IC80 ratio of COX-1:COX-2, and pharmacodynamic parameters, such as the median effective concentration, median effective dose, and maximal effect response, can then be calculated [104,105,106].

Acute-phase proteins are frequently used to evaluate the efficacy of NSAIDs in both orthopedic and infectious disease models in equine research. Serum Amyloid A (SAA) is a major acute-phase protein and α-globulin produced primarily by the liver in response to inflammatory stimuli and acts as both an immunomodulatory and pro-inflammatory protein [107]. SAA increases in response to inflammatory stimuli by 100–1000-fold, peaking at 48 h, before gradually declining to baseline (0.5–20 mg/L) following cessation of the inflammatory stimuli [107]. In horses, SAA has become increasingly popular as a stall-side diagnostic test, with many practitioners using it to rule out the presence of an infectious vs. non-infectious etiology, with a proposed cutoff value of 50 mg/L [108]. However, it is important to note that SAA is a measure of the acute-phase (i.e., inflammatory) response, rather than a measure of specific etiology. SAA has been found to significantly increased in the sterile inflammatory state provoked following administration of LPS (*E. coli* O55:B5) [78]. Following administration of LPS, SAA was greater than 1000 mg/L at 23 h post-infusion and remained elevated for greater than 72 h post-LPS infusion [78].

In differentiating between clinically sick horses and clinically normal horses, SAA was found to have a 75% diagnostic test accuracy, with a 53% sensitivity and 94% specificity [109]. When examining diagnostic tests, tests with high specificity and low sensitivity are useful at “ruling a diagnosis in” when a positive test result in achieved [110]. Additionally, when evaluating diagnostic tests, a cutoff value of 1.5 for sensitivity + specificity has been established as acceptable agreement for test utility, where 2 is equal to perfect agreement and 1 is equal to chance [110]. When examining the sensitivity and specificity of SAA in differentiating clinically normal from clinically abnormal horses, the sensitivity + specificity=1.47, which is just below the cutoff value for diagnostic utility [111]. SAA was not found to be useful as a prognostic indicator for horses with inflammatory conditions on admission to a tertiary referral hospital [111]. While SAA may not be as useful as a diagnostic test, it has been shown to have greater utility in serial analysis, where increasing SAA over the course of hospitalization was found to be associated with increased complication rates and recommendation of euthanasia [111]. Therefore, the utility of SAA as an outcome measure in pharmacodynamic and efficacy studies may be limited dependent on model selection and the duration of study sampling.

## 6. Conventional NSAIDs in Equine Practice

Conventional NSAIDs, such as the non-selective COX inhibitors phenylbutazone and flunixin meglumine, are the analgesic agents most widely used by US equine practitioners, accounting for 29.4% and 25.7% of NSAID prescriptions [1]. NSAIDs are commonly prescribed in equine practice for their anti-inflammatory, analgesic, and antipyretic properties, and conventional NSAIDs accomplish these goals through inhibition of COX-mediated prostaglandin production.

NSAIDs, such as phenylbutazone, flunixin, and ketoprofen, tend to have a duration of effective action beyond that suggested by their concentration vs. time profiles or their short elimination half-lives. This is largely due to accumulation of the drugs in inflammatory exudates. This accumulation is caused by a combination of the drugs’ high plasma protein binding (>98%) and relatively low pKa levels [112]. This contrasts with NSAIDs that have non-acidic functional groups (such as the non-conventional NSAID, acetaminophen), which are distributed throughout the body in a homogenous manner with minimal accumulation in tissues or inflammatory exudates. The inflammatory state is an acidic microenvironment that also results in an influx of albumin and other proteins into the inflamed tissue or fluid space. Therefore, when mildly acidic NSAIDs are present in an acidic inflammatory microenvironment, ion trapping of these drugs ultimately results as they are shifted into the intracellular space and cell membranes [113]. While the protein influx may carry the highly protein-bound drugs into the inflamed tissue, the acidic microenvironment inhibits protein binding, leading to further accumulation of free drug in the intracellular space [113].

Knowledge of the degree of COX-1 and COX-2 inhibition can be helpful when evaluating NSAIDs both from pharmacodynamic and safety standpoints. COX-1: COX-2 selectivity can be evaluated using the IC50 (50% of the maximal inhibitory concentration) or IC80 (80% of the maximal inhibitory concentration) ratio of COX-1: COX-2 as measured by their respective surrogate markers TXB_2_ and PGE_2_. While the IC50 has been traditionally used to evaluate the relative inhibition of NSAIDs on each COX isoform, the IC80 is more strongly associated with analgesic effects, as anti-inflammatory effects occur at 80% COX-2 inhibition [114]. Throughout this review, both IC50 and IC80 data will be reported when available. While in vitro quantification of COX-1 and COX-2 inhibition can assist in drug discovery, it is not always predictive of the degree of COX-1 vs. COX-2 selectivity in vivo. Therefore, it is advisable to view in vitro COX selectivity as a tendency rather than an absolute reflection of in vivo COX-selectivity—particularly under different dosing conditions.

### 6.1. Non-Selective Cyclooxygenase Inhibitors

#### 6.1.1. Pharmacokinetics and Efficacy of Phenylbutazone

Phenylbutazone is the most prescribed non-selective COX inhibitor for treatment of pain associated with musculoskeletal disorders and lameness in horses [1,115]. Phenylbutazone is a non-selective COX inhibitor and has an in vitro IC50 COX-1:COX-2 of 0.302 and an IC80 of 0.708 in equine whole blood [116]. Phenylbutazone is a member of the pyrazolidine drug class and is metabolized into two main active metabolites, oxyphenylbutazone and γ-hydroxyphenylbutazone. Phenylbutazone is highly protein-bound (98%), with a low volume of distribution (Vd; 0.17 L/kg), and therapeutic plasma concentrations are estimated to be between 1 and 4 µg/mL using PK/PD modelling [117].

Following intravenous administration of phenylbutazone at 3.5–4.5 mg/kg (2 g/horse), the plasma concentration was 42.3 µg/mL at 5 min post-administration, and the terminal elimination half-life was 13.9 h (Table 1) [118].

Following oral dosing, phenylbutazone has been found to bind to digesta in horses, which may result in prolonged absorption and time to effect, with a Tmax of 3.8 h and bioavailability of 78% in fasted ponies vs. a Tmax of 13.2 h and bioavailability of 69% in ponies with access to free-choice feed (Table 1) [119]. This has led to dual-phase absorption, with primary absorption of initial free drug from the proximal small intestine, followed by a secondary absorption phase once the drug has been liberated from the digesta in the large intestine [120]. Additionally, there have been differences in the bioavailability of different formulations of phenylbutazone reported in fasted horses, with a mean bioavailability of 74.1% for the tablet formulation and 87.0% for the paste formulation (Table 1) [121].
animals-13-01597-t001_Table 1Table 1Selected pharmacokinetic parameters for phenylbutazone in the adult horse.DrugFormulationDose(mg/kg)No. of DosesRouteCmax(µg/mL)Tmax(h)T ½(h)F(%)NotesRef.PhenylbutazoneInjectable3.5–4.51IV

13.9100
[118]PhenylbutazoneInjectable41IV

10.9 ± 5.32 100Fasted horses[121]PhenylbutazoneInjectable4.41IV

5.02100Fasted ponies[119]PhenylbutazoneInjectable4.41IV

4.71100Fed ponies[119]PhenylbutazonePowder4.41PO11.8 ± 2.35.9 ± 1.85.677 ± 10Fasted ponies[119] PhenylbutazonePowder4.4 1PO11.9 ± 1.113.2 ± 1.25.669 ± 5Fed ponies[119] PhenylbutazonePaste41PO12.2 ± 4.02.78 ±0.5713.4 ± 3.0187.0Fasted horses[121]PhenylbutazoneTablet41PO10.5 ± 4.254.72 ± 4.1415.1 ± 3.9674.1Fasted horses[121]Cmax: maximum plasma concentration; Tmax: time to reach maximum plasma concentration; T ½: elimination half-life; F: bioavailability; Ref.: reference.


Phenylbutazone, when given intravenously at 4.4 mg/kg, has been found to be effective in improving subjective lameness scores for 2–8 h post-treatment in horses with mechanically induced lameness when compared to placebo [58]. However, pharmacokinetic–pharmacodynamic (PK-PD) model simulations for phenylbutazone have predicted a maximal but transient analgesic effect for phenylbutazone administered intravenously at 1.5 mg/kg and a maximal analgesic effect persisting for 8 h for an intravenous administration of 2 mg/kg [104]. These doses would target an effective plasma concentration to elicit 50% of the maximal response (EC50) of 3.6 ± 2.2 µg/mL extrapolated from a study using horses with an induced arthritis model [105]. These dose simulations have led to the more current practice of a short course of therapy at 4 mg/kg followed by a dose reduction to 2 mg/kg for prolonged therapy to minimize adverse effects [113,115]. While phenylbutazone is an effective analgesic in lame horses, there have been conflicting results as to the effect of phenylbutazone on cartilage, and whether its administration is detrimental to cartilage healing and homeostasis. While an in vivo study suggested that phenylbutazone administration led to inhibition of proteoglycan synthesis in equine articular cartilage [122], these findings have not been corroborated by in vitro studies [123,124]. Therefore, while phenylbutazone administration may not have a deleterious effect on cartilage homeostasis in horses, there is no conclusive proof of any benefit of its administration on inflammation-associated cartilage catabolism despite improvement in synovitis-associated clinical signs [67].

Phenylbutazone is widely considered to be an antipyretic in horses by virtue of its global effects on prostaglandin inhibition; however, there is extremely limited research on its antipyretic efficacy or anti-inflammatory effects in equine models of SIRS [125]. Additionally, the effective plasma concentration for phenylbutazone for antipyresis in the horse is unknown. In ponies pre-treated with phenylbutazone (4.4 mg/kg IV) or flunixin meglumine (1.1 mg/kg IV) prior to LPS infusion (0.1 µg/kg), both drugs led to a less profound increase in heart rate and lactate than in untreated animals; however, there was no significant impact on rectal temperature [126]. The latter study determined that phenylbutazone was superior in blocking the LPS-mediated effects on bowel motility and that flunixin meglumine was more effective in blocking the cardiovascular effects of endotoxin [126]. In horses pre-treated with phenylbutazone (2 mg/kg) prior to LPS administration, phenylbutazone was found to result in a significant early decrease in TXB2 and attenuation (but not prevention) of the effects of endotoxemia [127]. Since pre-treatment studies are poor models for clinical disease—as practitioners are rarely able to pre-emptively treat SIRS and endotoxemia—further research is warranted.

#### 6.1.2. Pharmacokinetics and Efficacy of Flunixin Meglumine

Flunixin meglumine, a carboxylic acid drug, is a non-selective COX inhibitor with an in vitro IC50 COX-1:COX-2 of 0.336 and an IC80 COX-1:COX-2 of 0.436 in equine whole blood [116]. This finding has been corroborated in vivo, with a 168 h duration of thromboxane B2 suppression, a COX-1 proxy, as opposed to a 24 h duration of PGE_2_ suppression, a COX-2 proxy, for flunixin meglumine given IV at 1.1 mg/kg in healthy horses [118]. In the US, flunixin meglumine is currently FDA-approved for use in an injectable or in a paste or granule formulation for oral administration. Like phenylbutazone, administration of flunixin meglumine orally leads to alterations in drug pharmacokinetics. Therefore, selection of the formulation and the route of administration may lead to changes in the therapeutic response rate. Like phenylbutazone, oral administration of flunixin meglumine paste in the fed state has also been shown to decrease the Cmax (2.8 vs. 1.3 µg/mL) and increase the Tmax (0.76 vs. 7.66 h) without significant changes to total drug exposure as described by the area under the plasma concentration vs. time curve (AUC) in horses (Table 2) [128]. The decreased C_max_ and increased T_max_ are likely associated with alterations in gastrointestinal motility, gastric emptying rate, alterations in gastric pH with feeding, and binding of the free drug to feedstuffs [128]. Similarly, administration of the injectable preparation orally results in a 71.9% mean bioavailability (Table 2) [129]. These pharmacokinetic alterations are significant for the perceived therapeutic efficacy of the drug, as owners are unlikely to fast horses prior to medication administration and the most frequent systemic route of administration with the greatest compliance by owners and laypeople is via oral administration [4].

The pharmacokinetics and pharmacodynamics of an FDA-approved transdermal formulation of flunixin meglumine for cattle have been described in horses. Following application of transdermal flunixin meglumine at an average dose of 0.88 mg/kg, the average Cmax was 0.52 µg/mL within 8.67 h, with an elimination half-life of 24.3 h (Table 2) [130]. In comparison to other routes of administration, the Cmax was much lower and the Tmax was much later for transdermal flunixin [130] than for the oral administration (5.3 µg/mL, 0.71 h) (Table 2) [128] or intravenous administration (19.8 µg/mL, instantaneously) [131]. Despite the alterations in pharmacokinetics, the duration of TXB2 (COX-1) and PGE2 (COX-2) inhibition for transdermal flunixin meglumine was similar to that of IV flunixin meglumine, lasting for up to 168 h for TXB2 and 24 h for PGE_2_ [118,130].

**Table 2 animals-13-01597-t002:** Selected pharmacokinetic parameters for flunixin meglumine in the adult horse.

Drug	Formulation	Dose (mg/kg)	No. of Doses	Route	Cmax or C_0_ (µg/mL)	Tmax (h)	T ½ (h)	F (%)	Notes	Ref.
Flunixin meglumine	Injectable	1.1	1	IV	19.8 ± 2.7		3.38 ± 1.14	100	Horses	[131]
Flunixin meglumine	Injectable	1.1	1	IV	19.1 ± 3.4		2.96 ± 1.00	100	Miniature horses	[131]
Flunixin meglumine	Injectable	0.85–1.1	1	IV			9.68 ± 6.82	100	Fasted horses	[118]
Flunixin meglumine	Injectable	1.1	1	IV			2.1	100	Fed horses	[128]
Flunixin meglumine	Injectable	1.1	1	PO	5.27 ± 1.93	0.71 ± 0.1	2.8	71.9 ± 26.0	Fed horses	[129]
Flunixin meglumine	Paste	1.1	1	PO	1.31 ± 0.23	7.66 ± 1.74	2.9		Fed ponies	[128]
Flunixin meglumine	Paste	1.1	1	PO	2.84 ± 0.28	0.76 ± 0.17	2.8		Fasted ponies	[128]
Flunixin meglumine	TD	0.88	1	TD	0.515 (0.37–0.71)	8.67 (8.0–12.0)	22.4 (18.3–42.5)			[130]

Cmax: maximum plasma concentration; Tmax: time to reach maximum plasma concentration; T ½: elimination half-life; F: bioavailability; TD: transdermal.

Classically, veterinarians have preferred phenylbutazone for the treatment of somatic pain and flunixin meglumine for the treatment of visceral pain [1]. However, flunixin meglumine has been shown to be equally effective in reducing lameness in horses with navicular syndrome as measured by force-plate analysis [132]. Flunixin meglumine has also been found to be effective in reducing subjective lameness scores in horses with mechanically induced lameness for up to 12 h post-treatment (1.1 mg/kg IV) [59]. However, the reduction in lameness was found to be more rapid following IV phenylbutazone administration than flunixin meglumine [59]. In contrast, flunixin meglumine (1.1 mg/kg IV) was found to have no greater reduction in both subjective and objective lameness scores when compared to placebo in a separate study utilizing a mechanically induced lameness model [133]. However, it should be noted that the horses in this study only achieved plasma concentrations above the EC50 of 0.93 µg/mL for flunixin meglumine for a maximum duration of 4 h post-administration [105,133]. This contrasts with previous studies which demonstrated maintenance of therapeutic plasma concentrations for up to 16 h following a single dose of 1.1 mg/kg IV flunixin meglumine [105]. Therefore, wide inter-individual variation in the pharmacokinetic disposition of flunixin meglumine may have played a role in the low therapeutic response rate [133].

Veterinarians consider flunixin meglumine the NSAID of choice for treating the clinical effects of endotoxemia [134]. In contrast to phenylbutazone, the effects of flunixin meglumine on pro-inflammatory prostanoids and cytokines in endotoxemic horses are well elucidated. Flunixin meglumine, when given at label dosing (1.1 mg/kg IV), has been shown to be effective in relieving the clinical signs and cardiopulmonary effects of LPS administration in horses [135,136,137]. A sub-label dose of flunixin has often been cited as being “anti-endotoxic” due to the results of a pair of studies from the 1980s, where administration at 0.25 mg/kg or 1 mg/kg IV prior to LPS administration resulted in significant reduction in TXB2 and prostacyclin [138]. However, only the 1 mg/kg dose of flunixin meglumine ameliorated the clinical signs of endotoxin administration, and horses treated with 0.25 mg/kg flunixin meglumine still developed profound tachycardia, tachypnea, colic, and pyrexia [138]. When low-dose flunixin meglumine (0.25 mg/kg IV q8hr) was administered after LPS administration, flunixin meglumine-treated horses had less profound leukopenia and lower TXB2 concentrations than untreated horses [139].

These studies demonstrated that treatment with flunixin meglumine results in reduced prostanoid production in response to endotoxin. However, a drug cannot be considered disease-modifying, or “anti-endotoxic,” without directly counteracting the initial profound pro-inflammatory NF-kB-mediated cytokine release resulting from the interaction between LPS, LPS binding protein, CD14, and TLR4. As the knowledge base surrounding the pathophysiology of endotoxemia has grown, subsequent studies have examined the effects of flunixin meglumine on cytokine release. While horses administered flunixin meglumine IV at 1.1 mg/kg were shown to have a reduced pyretic response, there was no significant effect on the levels of circulating TNF-α when compared to controls [140]. Additionally, pre-treatment with flunixin meglumine (1.1 mg/kg) prior to LPS administration resulted in a significantly lower white blood cell count and a trend toward higher TNF-α and IL-6 activity than the placebo [141]. However, in vitro work has demonstrated that flunixin meglumine (but not phenylbutazone) has inhibitory effects on NF-kB activation [142]. Furthermore, in vivo work has demonstrated that flunixin meglumine is an inhibitor of the matrix metalloproteinases MMP-2 and MMP-9, which are involved in the pathophysiology of laminitis [143]. Therefore, while flunixin meglumine is an excellent symptom-modifying agent (i.e., reduces clinical signs) in the treatment of endotoxemia in horses, its effects as a disease-modifying agent (i.e., anti-endotoxic or modifying inflammatory cytokine release) in vivo remain incompletely understood. Additionally, caution should be exercised in horses with pre-existing intestinal lesions administered flunixin meglumine, as it has been shown to increase permeability to LPS and impair mucosal healing in the small intestine [144].

#### 6.1.3. Pharmacokinetics and Efficacy of Ketoprofen

Ketoprofen is another non-selective COX inhibitor labelled in the US for IV administration (2.2 mg/kg once daily) for the treatment of inflammation and pain associated with musculoskeletal disorders in horses. Despite its widespread availability and multiple generic formulations, it is not widely used—accounting for only 4.6% of NSAID prescriptions in the US and <1% of NSAID prescriptions in the UK [1]. Ketoprofen, a 2-arylpropionic acid derivative, is available as a racemic mixture of its biologically inactive R(−) and biologically active S(+) enantiomers [145]. The S(+) enantiomer has been shown to be the predominant enantiomer in both exudates and transudates, and, similar to flunixin meglumine, ketoprofen enantiomers are eliminated from exudates and transudates more slowly than their plasma elimination rates [146]. Ketoprofen has an in vitro IC50 ratio COX-1: COX-2 of 0.48–1.07 [35]. Ketoprofen has been found to have similar persistence of PGE_2_ and TXB2 inhibition to flunixin meglumine. While having a similar Cmax in exudate fluids, flunixin meglumine (0.019 ± 0.01 ug/mL) has been found to have a significantly lower exudate PGE_2_ EC50 compared to ketoprofen (0.057 ± 0.009 ug/mL) in horses [146]. The serum EC50 for TXB_2_ for ketoprofen is 0.061 ± 0.016 ug/mL [146].

One reason ketoprofen may have fallen out of favor with practitioners is the uncertainty regarding its safety profile. One study determined that ketoprofen had less ulcerogenic and renal effects than the other non-selective COX inhibitors phenylbutazone and flunixin meglumine [147]. Ketoprofen, when given at an equimolar dose (3.63 mg/kg) to phenylbutazone (4.4 mg/kg) intravenously in horses with naturally occurring chronic laminitis, was found to be equally efficacious in reducing lameness scores and other pain metrics [148]. However, it should be noted that this dose is 1.65× times the recommended therapeutic dose of 2.2 mg/kg, and dosing ketoprofen at 2.2 mg/kg did not produce the same effect [148]. In an acute synovitis model, the synovial fluid area under the curve (AUC) for ketoprofen was significantly higher for horses with synovitis than for horses without synovitis, demonstrating its ability to sequester in the inflamed joint [145]. In horses undergoing routine castration, ketoprofen was found to have a similar analgesic efficacy to meloxicam and flunixin meglumine; however, it was noted that ketoprofen had a significantly less profound decrease in 24 h post-operative pain score than flunixin meglumine or meloxicam [149]. Additionally, one horse treated with ketoprofen presented with colic signs following castration, which was responsive to flunixin meglumine administration [149]. While no studies have been performed on the antipyretic efficacy of ketoprofen in horses, it is assumed to be similar to other non-selective COX inhibitors. In swine, pre-treatment with ketoprofen prevented an increase in rectal temperature in response to intravenous LPS, where acetaminophen and aspirin were not effective [150]. Post-LPS administration treatment with injectable ketoprofen led to a quicker decrease in rectal temperature in pigs when compared to acetaminophen and aspirin [150].

There is no FDA-approved oral formulation for ketoprofen in the US; however, a study examined the oral administration of both the IV formulation and a compounded paste formulation [151]. The IV formulation had a mean bioavailability of 69.5% for the R(−) enantiomer and of 88.2% for the S(+) enantiomer, while the bioavailability for the compounded paste formulation was 53% and 53% for the R(−) and S(+) enantiomers, respectively, in fasted horses (Table 3) [151]. However, when a different compounded paste with an oil base was administered orally to fed horses, ketoprofen could not be detected in three-quarters of horses [152]. When the same oil-based compounded paste was administered to fasted horses, the mean bioavailability was 2.67% and 5.73% for the R(−) and S(+) enantiomers, respectively, suggesting that an oil-based product preparation limits dissolution, and therefore absorption, of the drug in the gastrointestinal tract (Table 3) [152]. Oral administration of ketoprofen has been shown to increase the drug half-life and detection time for both plasma and urine of both enantiomers; however, no efficacy or pharmacodynamic studies have been performed for oral ketoprofen to determine its clinical utility [151].

## 7. The Use of COX-2 Selective Inhibitors in the Horse

Due to the detrimental side effects of COX-1 inhibition, a COX-2 selective inhibitor, firocoxib, has been developed and marketed for equine use, and other COX-2 selective agents have been used in an extra-label fashion in the US. The selectivity of COX-2 inhibitors is derived from their structure. They have been developed to fit the larger, hydrophobic binding site present on COX-2 and thus are size-limited from binding to the more linear binding site present on COX-1 [103]. However, it should be noted that the selectivity of COX-2 preferential inhibitors is concentration-dependent, and administration at supratherapeutic doses may result in COX-1 inhibition for some drugs [153].

### 7.1. COX-2 Selective Inhibitors

#### 7.1.1. Pharmacokinetics and Efficacy of Firocoxib in Horses

Firocoxib, a member of the coxib class, is classified as a highly selective COX-2 inhibitor with an in vitro IC50 COX-1:COX2 ratio of 263–643 in horses [154]. Compared to other NSAIDs, it is more lipophilic and relatively more non-ionizable, resulting in a longer plasma half-life (29.6 h for PO; 33.8 h for IV), a higher volume of distribution (1.5 L/kg), and greater tissue penetration and persistence [154]. Oral administration of firocoxib paste at the recommended label dose of 0.1 mg/kg resulted in complete bioavailability, with a Cmax of 74.2 ng/mL and a Tmax of 1.2 h [155]. Meanwhile, oral administration of the tablet formulation of firocoxib at 0.1 mg/kg resulted in an 87.8% bioavailability, with a significantly lower Cmax (57.92 ng/mL) and a significantly longer Tmax (3.2 h) (Table 4) [155]. However, it should be noted that horses were fasted prior to drug administration in this study, and therefore the bioavailability and maximum plasma concentrations attained may be altered in clinically treated animals [155]. Following administration of oral firocoxib at 0.1 mg/kg once daily for 14 days in horses where feed was not withheld, average plasma steady-state concentrations were reported as 94.9 and 112 ng/mL for the paste and tablet formulations, respectively [156]. Considering that pharmacodynamic modelling suggests an IC50 of 27 ng/mL and an IC80 of 108 ng/mL, due to the long half-life of the drug, a loading dose is advantageous to rapidly achieve plasma concentrations close to the IC80 following oral dosing [155]. Therefore, a single loading dose of 0.3 mg/kg orally prior to once-daily treatment at 0.1 mg/kg was investigated, and therapeutic concentrations were attained 1–3 days earlier than standard 0.1 mg/kg oral dosing (Table 4) [157].

Despite its ready availability in multiple formulations and the potential safety advantages of preferential COX-2 inhibition, firocoxib remains seldom prescribed in the US and UK, accounting for only 4.16% and 0.4% of NSAID prescriptions, respectively [1]. The reason for the low prescription rate may be a combination of cost and, due to its relatively new entry to the market, a lack of prescriber familiarity. While some clinicians anecdotally feel that the COX-2 selective inhibitors are less effective analgesics than the non-selective COX inhibitors, there are a number of studies that suggest equivalent analgesia between the two classes. In a large-scale field trial of firocoxib in horses with naturally occurring OA, there was equivalent overall clinical improvement between firocoxib paste (0.1 mg/kg PO) and phenylbutazone paste (4.4 mg/kg PO) following 14 days of once-daily treatment, with firocoxib demonstrating greater improvement in joint circumference, pain on palpation, and joint manipulation than phenylbutazone [159]. In a similar large-scale trial of horses with OA in single or multiple joints, firocoxib demonstrated significantly greater improvement in lameness scores following 14 days of treatment, with the greatest improvement noted in the first 7 days of treatment [160].

In post-operative horses recovering from a small intestinal strangulating obstruction, horses treated with firocoxib (0.3 mg/kg IV once, followed by 0.1 mg/kg IV q24h) were more likely to have a soluble CD14 (biomarker of endotoxemia) below the reference range when compared to horses treated with flunixin meglumine (1.1 mg/kg IV q12h) [161]. However, there was no significant difference in TNF-α between flunixin meglumine- and firocoxib-treated horses, and there were no significant differences in the development of SIRS, MODS, or clinical signs associated with either condition between groups [161]. In mares, pre-treatment with firocoxib prior to experimental induction of placentitis led to a decrease in allantoic (but not amniotic) PGE_2_, TNF-α, IL-6, IL-1β, IL-10, and PGF_2α_ when compared to untreated mares [162]. No prospective studies have been performed to evaluate the antipyretic efficacy of firocoxib in horses. In cats, pre-treatment with oral firocoxib (1.5 mg/kg) 14h prior to LPS administration more effectively attenuated the fever response in comparison to the non-selective COX inhibitor ketoprofen (2 mg/kg, oral) in an LPS model [163].

#### 7.1.2. Pharmacokinetics and Efficacy of Deracoxib in Horses

Deracoxib, like firocoxib, belongs to the coxib class of drugs; however, deracoxib is less COX-2-selective, with an in vitro IC50 COX-1:COX-2 of 25.67 and an IC80 COX-1:COX-2 of 22.06 in horses [164]. Deracoxib is FDA-approved for use in dogs for the treatment of post-operative pain and inflammation at a dose of up to 2 mg/kg/day for up to 3 days. Following oral administration of deracoxib at 2 mg/kg in horses, the Cmax was 0.54 µg/mL within 6.3 h, with a relatively long elimination half-life of 12.49 h [164]. In dogs, deracoxib has been associated with gastric perforation at higher doses or when combined with other NSAIDs or corticosteroids [165]. No efficacy or safety studies have been performed in horses.

#### 7.1.3. Pharmacokinetics and Efficacy of Meloxicam in Horses

While meloxicam is approved for use in horses in the UK, EU, Australia, and New Zealand, its use in the US remains extra-label. In the US, meloxicam accounts for <1% of NSAID prescriptions [1]. Despite its approval in the UK, meloxicam remains an unpopular NSAID amongst UK equine veterinarians, accounting for 1.4% of NSAID prescriptions [1]. Following a single oral dose of meloxicam at 0.6 mg/kg, the mean Cmax was 915 ng/mL, with a Tmax of 2.62 h and a mean elimination half-life of 10.24 h (Table 5) [166]. Daily dosing of meloxicam at 0.6 mg/kg for 6 weeks did not result in appreciable drug accumulation (AR = 1.11) [166]. In contrast to other NSAIDs, feeding status does not appear to significantly affect the pharmacokinetics of meloxicam in horses for a variety of oral preparations, with mean bioavailability ranging from 75 to 110% (Table 5) [167]. Protein binding is similarly high to other NSAIDs, with a mean plasma protein binding of 97.75% [167].

One potential reason for the low usage of meloxicam is the perceived lack of efficacy at the UK-approved once-daily dose of 0.6 mg/kg, despite integrated PK-PD modelling suggesting that this dose (the ED70) would be sufficient for lameness reduction based on a model of induced carpal arthritis [1,106]. In post-operative small intestinal strangulating obstruction cases, horses treated with meloxicam at a higher frequency than label indications (0.6 mg/kg IV q12h) had significantly higher pain scores and leukocyte counts when compared to horses treated with flunixin meglumine (1.1 mg/kg IV q12h) [170].

Additionally, in comparison to flunixin meglumine, horses treated with meloxicam had higher rectal temperature and more edema following castration with a greater acute-phase response than flunixin meglumine [171]. However, meloxicam when administered at 0.6 mg/kg has been shown to significantly reduce lameness and synovial fluid PGE_2_, substance P, and MMP in horses with LPS-induced synovitis at 8 and 24h when compared to placebo [172]. In horses with experimentally induced endotoxemia, oral meloxicam (0.6 mg/kg) resulted in a greater reduction in plasma TNF-α than oral flunixin meglumine (1.1 mg/kg) at 30, 60, and 90 min post-drug administration [68]. In the same study, body temperature was lower for flunixin meglumine at 3.5 and 4.5 h post-drug administration in comparison to oral meloxicam, while both drugs significantly reduced temperature in comparison to placebo for up to 6.5 h post-treatment administration [173].

A concern over the use of meloxicam is that, in contrast to the highly selective COX-2 inhibitor firocoxib, meloxicam is considered a COX-2 preferential inhibitor with an in vitro IC50 COX-1:COX-2 of 3.8 [116]. When the IC80 is considered, the COX-2 selectivity decreases, with an in vitro IC80 COX-1:COX-2 of 2.2 [116]. Therefore, some practitioners are concerned that the safety profile for meloxicam would not be significantly improved over non-selective COX inhibitors. However, meloxicam, when administered at the recommended dose (0.6 mg/kg PO once daily for 6 weeks), was not associated with any evidence of gastrointestinal, renal, clinicopathologic, or clinical toxicity [166]. When administered at 3× (1.8 mg/kg once daily) and 5× (3.0 mg/kg once daily) the recommended dosage for 6 weeks, horses developed dose-dependent evidence of NSAID toxicity, including gastric ulceration, hypoproteinemia, right dorsal colitis, nephrotoxicity, and death [166].

#### 7.1.4. Pharmacokinetics and Efficacy of Carprofen in Horses

Carprofen is a member of the carboxylic acid class of NSAIDs that is considered to be a COX-2 preferential inhibitor in horses, with an in vitro IC50 COX-1:COX-2 of 1.996; however, the COX-2 selectivity decreases with higher concentrations, as the IC80 COX-1:COX-2 is 1.74 [116]. Carprofen is supplied as a racemic mixture of its R(−) and S(+) enantiomers. Carprofen is only approved for use in dogs in the US but is approved for use in horses in the EU for treatment of musculoskeletal pain and post-operative inflammation. Despite its approval in the EU and Canada, carprofen remains an unpopular NSAID in horses worldwide, accounting for <1% of prescriptions in the US, Canada, and the EU [1].

Following administration of 0.7 mg/kg IV to horses, the average plasma concentration of carprofen was 12.61 µg/mL at 5 min post-administration, with a relatively long elimination half-life of 18.1 h (Table 6) [174]. In comparison to ketoprofen, carprofen has a much longer elimination half-life and lower clearance rate for both of its enantiomers [175]. The volume of distribution of carprofen is low at 0.25 L/kg; however, carprofen has still been found to penetrate inflammatory exudates and produces moderate inhibition of serum TXB2 and exudate PGE_2_ [174].

Following orthopedic surgery, horses treated with carprofen (0.7 mg/kg IV) required analgesia an average of 11.7 h post-operatively, which was significantly earlier than for flunixin meglumine (1 mg/kg IV, 12.8 h) but significantly longer than for phenylbutazone (4 mg/kg IV, 8.4 h) [177]. The effective plasma concentration for analgesia in horses treated with carprofen is 1.5 µg/mL, and carprofen at 0.7 mg/kg IV was found to have superior analgesia in response to thermal stimulus to flunixin meglumine for up to 24 h post-administration [178]. Administration of oral carprofen at 2× the recommended dose (1.4 mg/kg) orally for 14 days resulted in significant reductions in albumin, globulin, and total protein, as well as ventral edema [179]. There is a lack of any efficacy data on the use of carprofen in horses with naturally occurring or experimentally induced endotoxemia to determine its antipyretic effect in horses.

#### 7.1.5. Pharmacokinetics and Efficacy of Etodolac in Horses

Etodolac is a pyranoindole acetic acid derivative that is a COX-2 preferential inhibitor, with an in vitro IC50 COX-1:COX-2 of 4.32 and an IC80 COX-1:COX-2 of 4.77 [180]. Etodolac exists in a racemic mixture of both its R(−) and S(+) enantiomers, with its S(+) enantiomer causing its anti-inflammatory effects, while its R(−) enantiomer dominates in plasma due to chiral inversion [180]. Following IV administration of etodolac in horses at 20 mg/kg, the mean Cmax was calculated to be 91.44 µg/mL, and the mean elimination half-life was 2.67 h [180], whereas, following oral administration of etodolac in horses at 20 mg/kg, the mean Cmax was 32.57 µg/mL within 1.03 h, with an elimination half-life of 3.02 h and a bioavailability of 77.02% [180]. In comparison to other species, horses have much higher clearance (234.87 mL/kg/h) of etodolac [180].

Etodolac has been examined as an analgesic agent in horses with naturally occurring and induced lameness. In horses with navicular syndrome, administration of etodolac at 23 mg/kg orally every 24 h for 3 days resulted in improvement in lameness as evidenced by increased mean peak vertical force for up to 24 h post-treatment [181]. Increasing the frequency of drug administration to every 12 h for 3 days did not provide any further improvement in lameness, and no adverse effects were noted in either dosing group [181]. In horses with LPS-induced synovitis, etodolac (23 mg/kg IV q12h) equally prevented an increase in synovial fluid WBC count and PGE_2_ concentration when compared to phenylbutazone, while a greater number of phenylbutazone-treated horses (4/6) were free of lameness compared to those treated with etodolac (3/6) and controls (2/6) [182]. The safety and antipyretic efficacy of etodolac have not been examined in horses.

### 7.2. Adverse Effects of Conventional Non-Steroidal Anti-Inflammatory Drugs in Horses

As stated earlier, NSAIDs, such as phenylbutazone, flunixin, and ketoprofen, tend to have a duration of effective action beyond that suggested by their concentration vs. time profiles or their short elimination half-lives. This is largely due to accumulation of the drugs in inflammatory exudates, primarily a property of the drugs’ high plasma protein binding (>98%) and relatively low pKa levels [112]. However, these same properties that aid in the accumulation of acidic NSAIDs in inflammatory tissues—their effective sites of action—also lead to the accumulation of NSAIDs in sites that ultimately cause adverse effects.

Both non-selective COX and COX-2 selective inhibitors have been associated with an increased risk of squamous gastric ulcer formation in horses treated with a short course of oral therapy when compared to placebo [183]. COX-1 is expressed in most tissues, where it mainly functions to maintain a degree of basal prostanoid synthesis. COX-1 activity in the gastrointestinal tract is cytoprotective, where it is primarily responsible for the production of PGE_2_ and PGI_2_ [184]. Both of these prostanoids serve their cytoprotective functions by promoting the secretion of bicarbonate and mucous and decreasing gastric acid secretion in the GI tract to aid in the neutralization of acidic GI contents, as well as by promoting mucosal blood flow [184].

As weak organic acids, NSAIDs are non-ionized and relatively lipid-soluble within the acidic gastric fluid, leading to diffusion across the pH-neutral gastric mucosa [184]. Once these drugs have crossed into the neutral pH of the gastric mucosa, they become ionized and relatively lipophobic, leading to ion trapping and cellular injury through uncoupling of mitochondrial oxidative phosphorylation [184]. Additionally, suppression of PGE_2_ in the gastric mucosa may lead to changes in mucosal blood flow, causing the region to be more prone to ischemic injury; however, this theory has not been substantiated in vivo [185]. The uncoupling of mitochondrial oxidative phosphorylation paired with a decrease in mucosal blood flow leads to increased mucosal permeability and exposure of the gastric mucosa to caustic gastrointestinal contents, such as bile and gastric acid.

Since COX-2 inhibitors failed to produce gastric ulcers in humans, it was long postulated that the increase in gastric ulceration seen with non-selective COX-inhibitor administration was due to the effects of COX-1 inhibition [114]. In a safety study of repeated oral dosing of firocoxib, treatment with firocoxib produced significantly more squamous and glandular ulceration than placebo [183]. When compared to the non-selective COX inhibitor phenylbutazone, firocoxib induced less glandular ulceration and a similar degree of squamous ulceration, indicating that the NSAID-induced gastric ulceration is not solely dependent on COX-1 inhibition [183]. In horses, COX-1 has been found to be the predominant isoform expressed in squamous mucosa of the healthy stomach, while COX-2 expression was significantly increased compared to COX-1 expression in ulcerated mucosa, suggestive of COX-2′s role in healing gastric ulceration [186].

To optimize therapeutic outcomes while minimizing the risk of gastric ulceration, it would therefore be prudent to select the lowest effective dose of an NSAID with a relatively short plasma half-life to minimize the duration of exposure of the gastrointestinal tract to circulating drugs [114]. While NSAIDs have been shown to induce both squamous and glandular gastric ulceration in horses under experimental conditions, the actual correlation between NSAID use and gastric ulceration in the general horse population remains unproven, and therefore NSAID use should be considered a risk factor in the development of multifactorial disease [187,188].

While phenylbutazone is most strongly implicated, both non-selective COX inhibitors and COX-2 selective inhibitors have also been associated with the development of right dorsal colitis in horses [166,189]. While the root cause of lesion localization to the right dorsal colon in horses has not been elucidated, it may be a combination of an increased sensitivity of the right dorsal colon to alterations in intestinal blood flow, changes in the intestinal microbiome and volatile fatty acid production, and alterations in lipoxygenase mRNA expression [185].

Despite the high COX-2 selectivity of firocoxib, adverse effects can still occur, and it is not safe for co-administration with non-selective COX inhibitors. Following 10 days of administration of firocoxib (0.1 mg/kg) and phenylbutazone (2.2 mg/kg), there was a significant increase in serum creatinine and decrease in total protein levels [190]. While the coxib class of drugs have been linked to an increased risk of thrombotic events in humans, short-term treatment of healthy horses with firocoxib (0.3 mg/kg PO once, followed by 0.1 mg/kg PO q24h for 4 days) was not shown to alter viscoelastic coagulation profiles [191]. However, caution should still be exercised with critically ill horses or horses with evidence of hypercoagulability, as the safety of these drugs has not been examined in diseased animals.

The adverse renal effects of NSAID administration have been well documented in all species. Both COX-1 and COX-2 inhibition leads to reduction in the downstream production of prostaglandins, such as PGE_2_, PGD_2_, and prostacyclin, which are vital for renal homeostasis. Prostaglandins aid in the maintenance of renal blood flow through their vasodilatory effects on the afferent renal arteriole, acting to increase renal perfusion, decrease renal vascular resistance, and distribute blood flow to the renal medulla [80]. Therefore, when COX-inhibiting NSAIDs are administered, there is impairment of renal vasomodulation, which can lead to acute renal injury and renal medullary crest necrosis—particularly in disease states causing a systemic inflammatory response or in polytherapy with other nephrotoxic agents [192,193]. For further review of the adverse effects of NSAIDs in the horse, there are several excellent reviews on the topic [194,195,196].

## 8. Non-conventional NSAIDs in Horses

### 8.1. Pharmacokinetics and Efficacy of Dipyrone (Metamizole) in Horses

Dipyrone (metamizole) is a pyrazolone derivative that was recently approved for use in horses in the US following a previous withdrawal from the market due to risks to human health (agranulocytosis). Dipyrone was found to have low usage compared to other NSAIDs during the period of 1998–2013 (<1%) in the US; however, use is likely to be on the rise with its return to the market in an approved form [1]. Dipyrone itself is a prodrug which is rapidly hydrolyzed to 4-methyl-amino-antipyrine (4-MAA), which can subsequently be hepatically metabolized into three other major metabolites (4-amino-antipyrine (AA), 4-formyl-amino-antipyrine (FAA), and 4-acetyl-amino-antipyrine (AAA)) [197]. Dipyrone is FDA-approved for the control of pyrexia at a dose of 30 mg/kg IV once to twice daily for up to 3 days in horses. While dipyrone has analgesic and antipyretic effects, it is a weak anti-inflammatory, and the mechanism of its action remains incompletely understood, despite many years of clinical use. It has been proposed that dipyrone exerts its antipyretic effects through central COX inhibition and/or through interaction with TRPV1 or TRPA1 receptors [198]. While for many years it was postulated that both dipyrone and acetaminophen exerted their analgesic effects through interaction with central COX-3 (a splice variant of COX-1), this theory has largely been refuted, as a physiologically functional COX-3 has yet to be sequenced in humans and the experimental methodology used to prove the “COX-3 hypothesis” has been found lacking [199]. There is increasing evidence from a variety of models that both acetaminophen and dipyrone interact with the endocannabinoid system, as metabolites of both drugs have been shown to interact with the CB1 receptors, leading to desensitization of TRPV1 channels and analgesic effects [198,200].

When administered at 30 mg/kg IV to horses, dipyrone reaches a mean plasma concentration of 40.6 µg/mL at 15 min post-administration, has a mean elimination half-life of 4.5 h, and has minimal evidence of accumulation when administered twice daily for 9 days (AR = 1.19) (Table 7) [201]. However, accumulation does occur following IV administration of dipyrone at an increased frequency (30 mg/kg q8h; AR = 2.59) or increased dose (90 mg/kg q12h; AR = 9.27) (Table 7) [201]. Dipyrone is approximately 56% protein-bound in humans [202], while the volume of distribution in horses is 1.3 L/kg [203]. The high volume of distribution coupled with the relatively low plasma protein binding means that dipyrone is distributed throughout the body and that a larger fraction of free drug is available to cross the blood–brain barrier to its site of action in the central nervous system. Oral dipyrone is available in some countries; however, its pharmacokinetics and efficacy have not been reported in the horse.

Despite decades of use in equine practice, research on the pharmacodynamics, efficacy, and safety of dipyrone in horses is lacking. When administered to horses with naturally occurring fever, dipyrone demonstrated a significantly greater reduction in rectal temperature by 6 h post-administration of 30 mg/kg IV in comparison to placebo [205]. No clinically significant adverse events were noted following treatment with dipyrone in a clinical field trial (30 mg/kg IV up to every 8 h for up to eight doses) [205]. Dipyrone has been frequently used by equine practitioners for the treatment of colic, where, in combination with n-butylscopolammonium bromide, its mild analgesic effects were found to be favorable to prevent masking of conditions necessitating surgical correction [141]. Dipyrone, when combined with n-butylscopolammonium bromide, was found to be an effective analgesic in five ponies with experimentally induced visceral pain, while dipyrone alone was only effective in two out of five ponies [206]. In humans, dipyrone is considered to have an improved safety profile for the upper GI tract and kidney compared to conventional COX-inhibitors [207], and dipyrone combined with meloxicam was superior to either drug as monotherapy in dogs undergoing ovariohysterectomy [208]. However, the safety of dipyrone in combination with conventional COX-inhibitors remains unknown, and caution should be exercised when combining dipyrone with other NSAIDs until further research is performed.

### 8.2. Pharmacokinetics and Efficacy of Acetaminophen (Paracetamol) in Horses

Acetaminophen (paracetamol) is the most common first-line antipyretic in humans [209]. However, there are no label-approved formulations for use in veterinary species, and therefore its use is considered extra-label worldwide. Like dipyrone, while acetaminophen is known to be an antipyretic and analgesic with weak anti-inflammatory activity, the mechanism of action remains incompletely understood. There is evidence for several different central effects, including interactions with the serotonergic, opioid, nitric oxide, and cannabinoid pathways, as well as effects on prostaglandin production [200]. The variety of central effects, particularly its interaction with the cannabinoid system, has led to the hypothesis that acetaminophen may be useful in the treatment of chronic and neuropathic pain states; however, evidence is limited and poor in quality at this time [210]. The onset of antipyresis with acetaminophen (0.5–1 h) compared to conventional NSAIDs, such as ibuprofen (3 h), is faster than the induction rate of COX-2 (2–4 h), suggesting that acetaminophen acts through an alternate central mechanism to conventional COX inhibitors [211,212]. The rapid antipyretic effect of acetaminophen is particularly advantageous for clinical cases when compared to conventional NSAIDs, and acetaminophen has been shown to be equally effective as a prophylactic therapy and as a treatment in humans with febrile disorders [213]. Furthermore, acetaminophen alternated with the non-selective COX inhibitor ibuprofen was found to have improved efficacy in the treatment of refractory fevers in children compared to either drug alone [214]. Acetaminophen alternated with ibuprofen has also been found to mitigate discomfort associated with fever, despite fewer doses being given when compared to monotherapy [215], and combination products are available over the counter in the US and other countries.

The pharmacokinetics of oral acetaminophen in horses have been well described following a variety of doses and durations of treatment. Acetaminophen, in general, is variably and rapidly absorbed orally in horses, and absorption is rate-limited by gastric emptying [60,216]. In humans, administration of acetaminophen in fed subjects has been shown to increase the Tmax and decrease the Cmax when compared to fasted subjects [217]. There is much debate over the analgesic threshold plasma concentrations for acetaminophen in humans; the effective concentration that elicits 50% of the maximum drug response (EC50) is estimated to be between 15.2 and 16.55 µg/mL, with a proposed minimum therapeutic concentration for analgesia of 10 µg/mL and 5 µg/mL for antipyresis [218,219]. Extrapolating from two intravenous acetaminophen pharmacokinetic studies in horses administered 10 mg/kg, an effective plasma concentration for analgesia has been calculated to be between 8 and 12 µg/mL in horses, yet these calculations have not been confirmed with in vivo or in vitro methods [220,221]. Following administration of a single oral dose of 30 mg/kg in systemically healthy adult horses, the average Cmax ranges from 20.83 to 30.02 µg/mL, within an average of 0.4h, with a mean elimination half-life of 4.6–5.3h (Table 8) [60]. Acetaminophen is 49.02% protein-bound, with a Vd of 1.35 L/kg [222] in horses, allowing for free drug to readily access its effective site in the central nervous system. In contrast to dipyrone, acetaminophen is primarily administered via the oral route in horses and has high oral bioavailability (91%) [222]. Acetaminophen has also been found to penetrate the aqueous humor in horses, with a mean aqueous humor:serum acetaminophen concentration ratio of 44.9% following administration of 20 mg/kg PO twice daily for 3 days [223].

Acetaminophen was first reported as an effective adjunct treatment for laminitis in one pony [227] and as an analgesic agent alone or when combined with NSAIDs in a model of inducible foot pain when administered at 20 mg/kg [61,62]. A study on the pharmacokinetics of acetaminophen at 20 mg/kg orally in horses found that it reaches proposed human therapeutic concentrations for analgesia (10 ug/mL) after administration, but pharmacokinetic simulation suggested that a dosage of 30 mg/kg every 12 h would allow for longer maintenance of the proposed therapeutic concentration [216]. Recent research in adult horses with mechanically induced lameness demonstrated a significant reduction in subjective lameness scores at 2 and 4 h post-treatment for oral acetaminophen at 30 mg/kg when compared to placebo, and there were no statistically significant differences in lameness reduction compared to oral phenylbutazone [60]. In that same study, acetaminophen at 20 mg/kg orally did not result in a significant reduction in lameness compared to other treatments (acetaminophen 30 mg/kg, phenylbutazone 2.2 mg/kg, and placebo) [60]. A follow-up study in horses with naturally occurring lameness found a significant improvement in subjective lameness scores at 2 and 4 h post-treatment and a significant improvement in the total difference in head height BMIS parameter (HD_tot_; indicator of forelimb gait asymmetry) at 1 h post-treatment for acetaminophen administered at 30 mg/kg orally when compared to the untreated baseline [226]. Additionally, there was a significant reduction in hindlimb lameness (PD_max_) as quantified by a body mounted inertial sensor (BMIS) system at 1, 2, and 8 h post-treatment for acetaminophen (30 mg/kg PO)-treated horses with clinically significant hindlimb lameness when compared to the untreated baseline [226]. Both studies suggest that acetaminophen provides pain relief in horses with lameness; however, this pain relief is transient, and acetaminophen is unlikely to be suitable for monotherapy in horses with moderate-to-severe orthopedic pain.

Oral acetaminophen has been reported to be effective as an antipyretic in an LPS-induced fever model in swine, though it was shown to not be as effective as oral ketoprofen [150]. However, horses—in contrast to swine—possess a lower first-pass effect for acetaminophen, which may allow for the drug to be more clinically effective [220]. A study was performed to determine the antipyretic efficacy and pharmacokinetics of acetaminophen in adult horses with experimentally induced endotoxemia, which demonstrated that acetaminophen was superior to placebo and not statistically different from the non-selective COX inhibitor flunixin meglumine with respect to fever reduction [225]. While both acetaminophen and flunixin meglumine significantly decreased rectal temperature at 4 and 6 h post-drug administration, flunixin meglumine demonstrated a greater post-treatment heart rate reduction than acetaminophen at 4 and 6 h [225]. In horses with experimentally induced endotoxemia, the pharmacokinetics of acetaminophen were altered in comparison to previously reported data in healthy horses. In healthy horses administered acetaminophen at 30 mg/kg orally, the mean Cmax was 20.83–30.02 µg/mL within a mean of 0.4 h after administration [60,226], while in endotoxemic horses the mean Cmax was 13.97 µg/mL within a mean of 0.6 h after administration of a 30 mg/kg oral dose (Table 8) [225]. Additionally, the AUC_0–8h_ was 44.8 h*µg/mL (range: 38.0–60.3 h*µg/mL) in endotoxemic horses [225], while the AUC_0–8h_ in healthy horses was 128.04 h*µg/mL (range: 71.9–176.2 h*µg/mL) [60]—indicative of lower total body exposure to the drug in endotoxemic horses.

The safety margin of acetaminophen is exceedingly wide, with therapeutic plasma concentrations found to be between 5 and 20 µg/mL at a dose of 50 mg/kg/day and the hepatotoxicity threshold at a plasma concentration of 150 ug/mL at a dose of greater than 150 mg/kg/day in human adults [228]. While acetaminophen carries a reputation for hepatotoxicity, acetaminophen-induced hepatotoxicity was not reported in a prospective clinical trial of over 30,000 human patients at therapeutic doses [229]. Some of the historical reluctance to use acetaminophen in large animals is also due to its safety profile in small animals. Toxicity of acetaminophen is well recognized in small-animal medicine, with over 1000 cases reported to the National Animal Poison Control Center in a 2-year period [230]. Clinical signs of APAP toxicity in small animals include methemoglobinemia and hemolysis (which predominate in cats) and hepatotoxicity in dogs [230,231,232]. Given that acetaminophen has demonstrated COX enzyme inhibition, there are similar concerns to conventional NSAIDs regarding its gastrointestinal safety profiles. Acetaminophen does not have any reported gastrointestinal side effects at recommended therapeutic dosing ranges in humans, and, due to its reported safety, acetaminophen has been recommended for use in humans with peptic ulcers as an analgesic [233]. Acetaminophen has also not been associated with any of the cardiovascular or renal side effects that are present with other COX-2 inhibitors [234].

In adult horses administered 20 mg/kg acetaminophen per os every 12 h for 14 days, there were no statistically significant differences in squamous or glandular gastric ulcer scores [216]. Additionally, there were no clinically significant alterations in clinicopathologic parameters, including liver-specific parameters, such as sorbitol dehydrogenase and bile acids [216]. In a separate population of 12 healthy adult horses administered 30 mg/kg of acetaminophen orally twice daily for 21 days, no statistically significant changes were noted following treatment in gastroscopy or liver biopsy scores, nor any significant changes in hepatobiliary enzymes when compared to pre-treatment values [226]. Therefore, acetaminophen is considered safe for use in healthy adult horses at doses of up to 30 mg/kg every 12 h for at least 21 days. Additionally, while acetaminophen has been combined or alternated with conventional COX-inhibitors in humans without evidence of toxicity [214], and anecdotally in horses, there are no studies investigating the safety or efficacy of combination or alternating therapy, and caution should be exercised when pursuing this treatment regimen.

### 8.3. Pharmacokinetics and Efficacy of Grapiprant in Horses

Grapiprant is a highly selective EP4 inhibitor of the piprant class, which prevents the binding of PGE_2_ to EP4 and thus inhibits the development of PGE_2_ mediated inflammation [235]. Grapiprant has been approved in the US for use in dogs at a dose of 2 mg/kg once daily to treat pain and inflammation associated with osteoarthritis [235]. When administered to horses at 2 mg/kg via nasogastric intubation, the mean Cmax achieved was 106 ng/mL within 30 min of administration, and plasma concentrations were below the limit of quantification (50 ng/mL) by 2 h post-administration [236]. However, when administered orally at 2 mg/kg via dosing syringe, the mean Cmax was 31.9 ng/mL within a Tmax of 1.5 h, and grapiprant was detectable in plasma for up to 72 h post-administration using more sensitive analytical methods (LOD: 0.005 ng/mL) [237]. Since the effective plasma concentration for analgesia in dogs has been determined to be 114–164 ng/mL, higher doses would likely be required in horses to achieve therapeutic effect [238]. Following a single oral administration of 15 mg/kg, grapiprant reached a mean Cmax of 327.5 ng/mL within a mean Tmax of 1 h, and a mean elimination half-life of 11.1 h was calculated [239]. Despite the prolonged elimination half-life of grapiprant, TNF-α stimulation was only noted for 2–4 h post drug administration, indicating that the pharmacodynamic effect of grapiprant is relatively short in the horse [239].

### 8.4. Cannabidiol in Horses

Cannabidiol (CBD) is becoming increasingly available and attractive to horse owners seeking alternative analgesics to conventional COX-inhibitors, despite a lack of efficacy and safety information or label approvals in veterinary species [240]. Cannabidiol primarily exerts its effects through its interaction with the endocannabinoid system. Though it has been postulated that CBD may also act as a COX-inhibitor to exert its anti-inflammatory activity, no evidence has been found to support this hypothesis thus far, and research has indicated that at low plasma concentrations CBD may produce pro-inflammatory cytokines in horses [240].

## 9. Conclusions

Both osteoarthritis and endotoxemia are associated with significant morbidity in horses, and treatment and management of these diseases has a significant economic impact on horse owners. The use of NSAIDs is widespread in horses, with the most-prescribed NSAIDs being non-selective COX inhibitors despite evidence of several adverse effects associated with their use. While COX-2 selective and preferential inhibitors have been introduced to the equine market and have a more favorable safety profile, their uptake has been slow amongst equine practitioners due to perceived lack of therapeutic efficacy, which has not been substantiated in the literature. There are a number of non-conventional NSAIDs that may be promising in the treatment of pain and inflammation in the horse, including acetaminophen and metamizole, that have an improved safety profile compared to conventional COX-inhibiting NSAIDs. Critical evaluation of the pharmacokinetics, pharmacodynamics, safety, and research methods of studies examining NSAID use in horses is critical for appropriate therapeutic selection.

## Figures and Tables

**Figure 1 animals-13-01597-f001:**
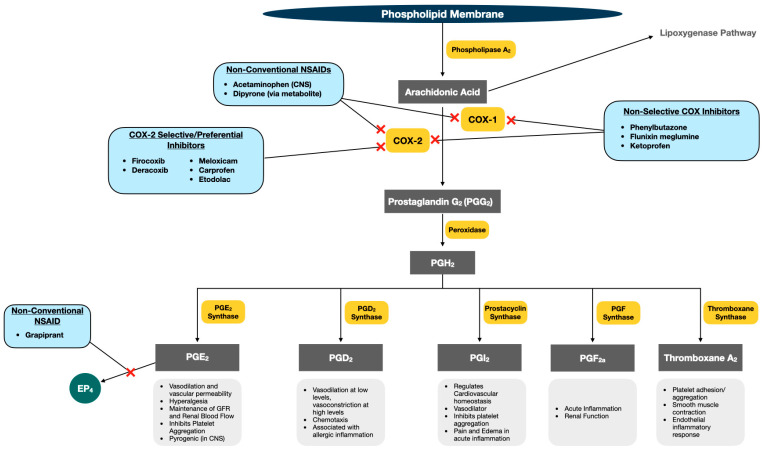
The arachidonic acid cascade and sites of NSAID action in the horse.

**Table 3 animals-13-01597-t003:** Selected pharmacokinetic parameters for ketoprofen in the adult horse.

Drug	Formulation	Dose (mg/kg)	No. of Doses	Route	Cmax (ug/mL)	Tmax (h)	T ½ (h)	F (%)	Notes	Ref.
Ketoprofen R(−)	Injectable	2.2 mg/kg	1	IV			0.70 ± 0.13	100	Fasted horses	[152]
Ketoprofen S(+)	Injectable	2.2 mg/kg	1	IV			0.99 ± 0.14	100	Fasted horses	[152]
Ketoprofen R(−)	Injectable	2.2 mg/kg	1	IV			2.49 ± 0.077	100	Fasted horses	[151]
Ketoprofen S(+)	Injectable	2.2 mg/kg	1	IV			2.86 ± 0.102	100	Fasted horses	[151]
Ketoprofen R(−)	Injectable	2.2 mg/kg	1	PO	1.2 ± 0.54	0.34 ± 0.05	5.67 ± 0.40	69.5 ± 10.3	Fasted horses	[151]
Ketoprofen S(+)	Injectable	2.2 mg/kg	1	PO	1.8 ± 0.56	0.55 ± 0.06	7.52 ± 0.37	88.2 ± 15.9	Fasted horses	[151]
Ketoprofen R(−)	Paste	2.0 mg/kg	1	PO	1.1 ± 0.38	0.52 ± 0.04	10.4 ± 1.59	65.9 ± 12.9	Fasted horses; compounded paste	[151]
Ketoprofen S(+)	Paste	2.0 mg/kg	1	PO	1.80 ± 0.69	0.84 ± 0.2	6.82 ± 0.91	65.2 ± 13.4	Fasted horses; compounded paste	[151]
Ketoprofen R(−)	Paste	2.2 mg/kg	1	PO	0.23 ± 0.03	0.5 ± 0.07	5.64 ± 0.60	2.67 ± 0.43	Fasted horses; oil-based paste	[152]
Ketoprofen S(+)	Paste	2.2 mg/kg	1	PO	0.28 ± 0.08	0.5 ± 0.07	5.69 ± 1.69	5.75 ± 1.48	Fasted horses; oil-based paste	[152]
Ketoprofen R(−)	Powder in gelatin capsule	2.2 mg/kg	1	PO	2.19 ± 0.44	0.83 ± 0.05	3.51 ± 0.99	50.50 ± 10.95	Fasted horses	[152]
Ketoprofen S(+)	Powder in gelatin capsule	2.2 mg/kg	1	PO	2.73 ± 0.49	0.92 ± 0.14	3.18 ± 0.93	54.17 ± 9.90	Fasted horses	[152]

Cmax: maximum plasma concentration; Tmax: time to reach maximum plasma concentration; T ½: elimination half-life; F: bioavailability; Ref.: reference.

**Table 4 animals-13-01597-t004:** Selected pharmacokinetic parameters for firocoxib in the adult horse.

Drug	Formulation	Dose (mg/kg)	No. of Doses	Route	Cmax or C_o_ (ug/mL)	Tmax (h)	T ½ (h)	F (%)	Accumulation Index	Notes	Reference
Firocoxib	Injectable	0.10	1	IV	0.21 ± 0.05		33.8 ± 11.2	100		2% Firocoxib in DMSO; fasted horses	[154]
Firocoxib	Injectable	0.10–0.134	1	IV	0.10 ± 0.015		31.07 ± 10.64	100		Commercial formulation; fasted horses	[155]
Firocoxib	Injectable	0.09	5	IV			39.36 ± 17.7	100	2.37 ± 0.24	Fasted horses	[156]
Firocoxib	Injectable	0.2	9	IV	1st: 0.31 ± 0.06 9th: 0.52 ± 0.13		44.2 ± 21.6	100	2.8 ± 1.1	Fed horses	[158]
Firocoxib	Paste	0.10	1	PO	0.075 ± 0.03	3.9 ± 4.4	29.6 ± 7.5	79 ± 31		Fasted horses	[154]
Firocoxib	Paste	0.10–0.134	1	PO	0.074 ± 0.02	1.2 ± 0.4	30.12 ± 5.85	111 ± 54.3		Fasted horses	[155]
Firocoxib	Paste	0.1	14	PO	1st: 0.03 ± 0.02 14th: 0.12 ± 0.04	0.52 ± 0.04	9.5 ± 8.9 316.8 ± 6.4		6.37 ± 7.56	Fed horses	[156]
Firocoxib	Paste	0.3 once followed by 0.1 for 9 doses	10	PO	0.18 ± 0.05	221.8 ± 5.3	41.76 ± 13.44			Fed horses	[157]
Firocoxib	Paste	0.1	12	PO	1st: 0.045 ± 0.01 12th: 0.173 ± 0.04	7.8 ± 4.8 0.79 ± 0.70	36.5 ± 9.5		3.8 ± 0.7	Fed horses	[158]
Firocoxib	Tablet	0.10–0.134	1	PO	0.058 ± 0.032	3.2 ± 1.09	32.77 ± 10.74	87.8 ± 54.3		Fasted horses	[155]
Firocoxib	Tablet	0.09 (average)	14	PO	1st: 0.043 ± 0.01 14th: 0.137 ± 0.05	6.5 ± 4.8 319.2 ± 5.5	41.5 ± 18.4		4.00 ± 1.05	Fed horses	[156]

Cmax: maximum plasma concentration; C_o_: initial plasma concentration; Tmax: time to reach maximum plasma concentration; T ½: elimination half-life; F: bioavailability.

**Table 5 animals-13-01597-t005:** Selected pharmacokinetic parameters for meloxicam in the adult horse.

Drug	Formulation	Dose (mg/kg)	No. of Doses	Route	Cmax or C_o_ (µg/mL)	Tmax (h)	T ½ (h)	F (%)	Notes	Ref.
Meloxicam	Injectable	0.6	1	IV			12.39 ± 4.07		Fasted horses	[167]
Meloxicam	Granule	0.6	1	NG	1.21 ± 0.32	1.5 ± 1.0	24.2 ± 3.73	110.37 ± 25.84	Fasted horses	[167]
Meloxicam	Granule	0.6	1	NG	0.85 ± 0.35	1.0 ± 0.25	34.08 ± 20.76	96.55 ± 46.96	Fed horses	[167]
Meloxicam	Suspension	0.6	1	NG	2.08 ± 0.64	1.0 ± 0.5	13.17 ± 5.25	88.3 ± 12.8	Fasted horses	[167]
Meloxicam	Suspension	0.6	1	NG	2.10 ± 0.84	0.5 ± 0.25	10.85 ± 6.31	75.4 ± 40.3	Fed horses	[167]
Meloxicam	Suspension	0.6	1	PO	0.915 ± 0.12	2.62 ± 1.88	10.24 ± 3.04		Fed horses	[166]
Meloxicam	Suspension	0.6	41	PO	1.01 ± 0.31		9.25 ± 2.64		Fed horses	[166]
Meloxicam	Suspension	0.6	1	PO	0.67 ± 0.196	5.5 ± 4.1	6.4 ± 3.0		Fasted horses	[168]
Meloxicam	Tablet	0.6	1	PO	0.708 ± 0.16	2.5 ± 0.8	6.5 ± 2.8		Fasted horses	[168]
Meloxicam	Tablet	0.6	1	PO	1.58 ± 0.71	3.48 ± 3.3	5.25 ± 1.4		Fed horses	[169]
Meloxicam	Tablet	0.6	14	PO	1.81 ± 0.76	1.93 ± 1.3	4.73 ± 1.3		Fed horses	[169]

Cmax: maximum plasma concentration; C_o_: initial plasma concentration; Tmax: time to reach maximum plasma concentration; T ½: elimination half-life; F: bioavailability; NG: nasogastric; Ref.: reference.

**Table 6 animals-13-01597-t006:** Selected pharmacokinetic parameters for carprofen in the adult horse.

Drug	Formulation	Dose (mg/kg)	Route	Cmax or C_o_ (ug/mL)	Tmax (h)	T ½ (h)	Notes	Ref.
Carprofen R(−)	Injectable	0.7	IV	8.36 ± 0.83		24.52 ± 3.39	Ponies	[176]
Carprofen S(+)	Injectable	0.7	IV	6.19 ± 0.34		7.95 ± 2.18	Ponies	[176]
Carprofen R(−)	Injectable	4	IV	38.39 ± 2.14		22.21 ± 1.37	Ponies	[176]
Carprofen S(+)	Injectable	4	IV	28.18 ± 1.76		11.35 ± 0.9	Ponies	[176]
Carprofen R(−)	Injectable	0.7	IV	5.54 ± 0.78		20.6 ± 2.55	Ponies	[121]
Carprofen S(+)	Injectable	0.7	IV	5.06 ± 0.92		16.8 ± 1.77	Ponies	[121]
Carprofen (total)	Injectable	0.7	IV	12.61 ± 1.43		18.1 ± 1.3	Ponies	[174]

Cmax: maximum plasma concentration; C_o_: initial plasma concentration; Tmax: time to reach maximum plasma concentration; T ½: elimination half-life.

**Table 7 animals-13-01597-t007:** Selected pharmacokinetic parameters for dipyrone and its metabolites in the adult horse.

Drug	Analyte	Formulation	Dose (mg/kg)	Freq.	No. of Doses	Route	Cmax or C_o_ (ug/mL)	Tmax (h)	T ½ (h)	AI	Notes	Ref.
Dipyrone	Dipyrone	Injectable	30	q12h	1	IV	40.6 ± 9.92		4.49 ± 0.67		Horses	[201]
Dipyrone	Dipyrone	Injectable	30	q12h	18	IV	48.5 ± 15.86		4.58 ± 1.23	1.19 ± 0.12	Horses	[201]
Dipyrone	Dipyrone	Injectable	30	q8h	1	IV	57.8 ± 6.48		3.58 ± 0.36		Horses	[201]
Dipyrone	Dipyrone	Injectable	30	q8h	27	IV	65.6 ± 7.84		5.77 ± 0.72	2.59 ± 0.24	Horses	[201]
Dipyrone	4-MAA	Injectable	25		1	IV			4.85 ± 5.04		Horses	[203]
Dipyrone	4-MAA	Injectable	25		1	IV	86.33 ± 36.55		3.34 ± 0.4		Horses	[204]
Dipyrone	4-MAA	Injectable	25		1	IM	24.14 ± 8.09	1.21 ± 0.56	3.00 ± 0.56		Horses	[204]

4-MAA: 4-methylaminoantipyrine; Freq.: frequency of administration; Cmax: maximum plasma concentration; C_o_: initial plasma concentration; Tmax: time to reach maximum plasma concentration; T ½: elimination half-life; AI: accumulation ratio; Ref.: reference.

**Table 8 animals-13-01597-t008:** Selected pharmacokinetic parameters for acetaminophen in the adult horse.

Drug	Formulation	Dose (mg/kg)	Freq.	No. of Doses	Route	Cmax or C_o_ (ug/mL)	Tmax (h)	T ½ (h)	Notes	Ref.
Acetaminophen	Injectable	10		1	IV			1.96 ± 0.47	Fed horses	[224]
Acetaminophen	Injectable	10		1	IV			2.41 ± 0.56	Fasted horses	[224]
Acetaminophen	Injectable	10		1	IV			4.30 ± 0.89	Fasted horses	[222]
Acetaminophen	Compounded solution	10		1	NG	14.44 ± 1.95	0.61 ± 0.27	3.97 ± 0.41	Fasted horses	[222]
Acetaminophen	Tablet	20		1	PO	16.61 ± 7.48	1.35 ± 1.69	2.78 ± 0.6	Fed horses	[216]
Acetaminophen	Tablet	20	q12h	28	PO	15.85 ± 6.64	0.99 ± 0.86	3.99 ± 0.69	Fed horses	[216]
Acetaminophen	Tablet	20		1	PO	20.01 (11.47–30.02)	0.66 (0.25–6)	3.5 (1.95–5.47)	Fed horses; induced lameness	[60]
Acetaminophen	Tablet	30		1	PO	30.02 (14.78–60.01)	0.43 (0.25–1)	5.3 (2.31–9.76)	Fed horses; induced lameness	[60]
Acetaminophen	Tablet	30		1	PO	13.97 (11.60–20.69)	0.65 (0.5–1.0)	3.11 (2.70–3.50)	Fed horses; endotoxemia model	[225]
Acetaminophen	Tablet	20	Twice daily	9	PO	18.64 ± 3.1	0.71 ± 0.62		Fed horses; dosing 7 h apart each day	[220]
Acetaminophen	Tablet	30	q12h	14	PO	20.83 ± 10.25	0.40 ± 0.22	2.95 ± 0.62	Fed horses; naturally occurring lameness	[226]
Acetaminophen	Tablet	30	q12h	42	PO	17.33 ± 6.91	0.67 ± 0.26	4.64 ± 3.56	Fed horses; naturally occurring lameness	[226]

Freq.: frequency of administration; Cmax: maximum plasma concentration; C_o_: initial plasma concentration; Tmax: time to reach maximum plasma concentration; T ½: elimination half-life.

## Data Availability

Not applicable.

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
