# Peer review of "The Clinical Pharmacology and Therapeutic Evaluation of Non-Steroidal Anti-Inflammatory Drugs in Adult Horses"

_animals, 2023, doi:10.3390/ani13101597_

Round 1

Reviewer 1 Report

The aim of this paper is to provide and overview of the clinical pharmacology for conventional and nonconventional NSAIDs (or drugs reported to potentially demonstrate some cox inhibition activity). The authors provide this overview in the context of treatment of two common disease processes (OA and inflammation associated with endotoxemia) for which NSAIDS are frequently administered. A fairly extensive overview is provided for conventional and nonconventional NSAIDs (including review of pharmacology and clinical efficacy) and extensive discussion is provided regarding experimental models and techniques used to evaluate the efficacy of NSAIDS. Overall, this is a well written and extensive review. This reviewer has provided some specific comments regarding wording, structure, and content of the manuscript.

I. Introduction:

1. Consider removing specific definitions of pharmacokinetic, pharmacodynamic and safety studies. Not sure this is needed or relevant to the manuscript. Instead may want to consider addressing why this paper fills a gap in knowledge of NSAIDS and why the authors have chosen to specifically focus on OA and inflammation associated with endotoxemia. Do these conditions represent the most common use of NSAIDs? It would be helpful in the introduction to also state which categories of NAIDS evaluated (e.g. conventional COX selective and non-selective; nonconventional….)

II. Disease Processes targeted by NSAIDS

1. Line 45: Consider stating “… for approximately 60% of lameness causes in adult horses”. Also this reference (3) is an AAEP proceeding. Consider providing the primary source for this statement.

2. Lines 54-76: this paragraph is quite long. Consider removing discussion of disease modifying drugs  (lines 54-58) and perhaps starting with “Disease modifying drugs are lacking for the treatment of OA in horses and thus treatment is limited to managing pain and inflammation associated with this disease through administration of symptom-modifying drugs. “

3. Lines 142-144: Should specifically mention how NSAIDs target COX pathway for management of LPS-mediated inflammation.

4. As a general comment, there are some inconsistencies in format of references.

5. Line 145 – consider replacing traditional with conventional in reference to these NSAIDS.

6. General comment: I recommend include a summary table listing drugs reviewed and how they are classified or primary mechanism of action.

7. General comment Lines 146-162 – consider adding a figure illustrating COX mediated pathway. The authors should consider briefly summarizing/defining COX-1 versus COX-2 activity (and what is known in the horse regarding these pathways) as this is emphasized in discussion of NSAIDS.

8. General comment paragraph Lines 221-248. This is a very long paragraph. Consider dividing this into 2 paragraphs. Perhaps at line 239 when begin discussion of transdermal flunixin.

9. Line 351 – subsection header 3.2 – this is unclear and placement of this discussion here in manuscript is confusing. It would make sense to either clarify that discussing only non-selective or both selective and non-selective NSAIDs or preferably to move this section to later in manuscript after the overview of the COX selective NSAIDS. As written the manuscript provides discussion of adverse effects associated with selective NSAIDS both here and in discussion of individual drugs but primarily addresses the adverse affects associated with nonselective NSAIDS in the section only.

10. General comment – The authors comment on effective plasma concentrations inconsistently in manuscript. When provide summary of reported pharmacokinetics should specifically comment whether or not effective plasma concentrations are known for drug for specific therapeutic purpose (e.g. analgesia, anti-inflammatory).

11. Line 422 – the section on Firocoxib discusses use as therapeutic for treatment of OA and as anti-pyretic agent. Please state whether this has been evaluated specifically for systemic anti-inflammatory properties in horses in response to LPS stimulation.

12. Line 545 – please clarify what type of surgery.

13. Line 578 – please consider changing title of section 5 to “Nonconventional” NSAIDS

14. For drugs covered in section 5 should consider organizing according to clinical relevancy or breadth of experimental knowledge associated with each drug, starting with the most clinical relevant or well studied (example acetaminophen).

15. Section 5.2 – Pharmacokinetics of Cannabidiol in horses – please consider removing or significantly shortening this section. While the use of CBD products has increased significantly in horses there is scant-to-no evidence regarding its actions as a COX-inhibitor. Discussion of CBD, although interesting, could be limited to lines 598-603 for purposes of this manuscript.

16. Line 755 – either specify which NSAIDs were compared to acetaminophen for these studies or specifically state conventional NSAIDs.

17. Lines 786-787 – specify how pharmacokinetics were altered in horses with experimentally induced endotoxemia. What was significance of this alteration. Also, cite which study referring to for healthy horses in this comparison.

18. Line 822 – consider renaming section 6 as “Experimental Models used to Evaluate Pain and Inflammation in Equine Research”. Both section 6 and section 7 are well written but as section headers are currently written it is unclear that one (section 6) is addressing the models used to create OA and LPS-mediated inflammation and the other is then addressing how the response to NSAID administration is evaluated in these (and other) experimental models or conditions.

19. Lines 1150-1153 – this sentence is very confusing as written. Please revise the grammar.

Reviewer 2 Report

This paper provides a review of NSAIDs in horses. When producing reviews, it is important that they be accurate because they are often read and referred to by practicing veterinarians who do not have the time, access to the primary literature, or necessarily the training to ensure the correctness of the material presented.

I think starting with the disease processes typically treated can work, but it is important to work in the information on cyclooxygenases and the two different enzymes (and their regulation) so that the reader has sufficient understanding before bringing up individual forms and use of terms like nonselective and selective COX inhibitors. I think better discussion of pain pathways is required, given that NSAIDs are used for main situations outside of the two identified where pain is involved. Probably useful early on to define eicosanoids and how they relate to prostaglandins and make sure to clearly differentiate PGE synthase from cyclooxygenases. This can then be useful when clarifying what COX enzymes actually produce and how this relates to the different PGs that are ultimately produced.

There is a tendency throughout the paper to refer to a lot of different papers without providing a clear, concluding synthesis from the available data. Doing so would improve the utility of this paper for its target audience.

Some discussion of spinal and CNS effects of NSAIDs, and related PK implications, would be useful to make the paper more complete. It helps to explain some of the potential differences between NSAIDs.

Over-all, this is a dense manuscript and would benefit from editing and reduction in length. Given that this is not a systematic review or a meta-analysis, mentioned so many papers in the text itself and summarizing the data makes it a long read.  Creating summarizing tables and just giving the conclusion in the text would make for an easier read.

Specific Points

1.       Definition of pharmacodynamics is incorrect. Given the important pharmacodynamic differences between NSAIDs, this needs to be defined correctly. Definition of PK could be improved.

2.       Line 48  “resulting in progressive”

3.       Description of peripheral sensitization and central sensitization lines 71 – 76 is simplistic and implies that peripheral sensitization is required for central sensitization, which is not true.

4.       Line 78 – spinal pathways are central nociceptive pathways.

5.       Description of endotoxemia (LPS-associated) without describing the different cyclooxygenase enzymes and the difference between COX and prostaglandin synthase enzymes will leave many readers confused. More information on the role of prostaglandins would be very helpful.

6.       Recommend moving up the description of COX-1 and COX-2

7.       Line 183:  bioavailability of different

8.       Line 233/234 – not clear what “these changes” are…..

9.       Flunixin section – describes different studies with markedly different PK values for oral flunixin – does not address these differences or explain what is the most relevant form.

10.   Paper in general does not describe the critical role of pharmacokinetic distribution of NSAIDs in inflamed tissue vs plasma, nor the importance of peripheral distribution of NSAIDs vs distribution into the CNS.

11.   The description of flunixin and endotoxemia and does not clearly address the issue of disease modification. I am not aware that there are any data that flunixin in horses changes endotoxemia outcomes.

12.   There seems in general to be a tendency to over-state the significance of minor differences in in vitro IC50 or IC80 ratios for NSAIDs and relate these to the in vivo situation.  Ultimately, selectivity is only relevant if it use in vivo does not inhibit COX-1 activity.  The dosing of all NSAIDs, including nonselective, has been determined based on anti-inflammatory and analgesic effects in vivo – in other words, enough to always inhibit COX-2 so at a tendency towards COX-1 selectivity is of no clinical significance. Both are inhibited at therapeutic doses of nonselective NSAIDs. And if a selective NSAID is used at a dose that inhibits COX-1 in vivo, it does not matter what its apparent in vitro selectivity is.

13.   Adverse effects section starts with PK discussion. PK discussion should proceed the individual drugs discussion as it is really important to understanding how they work.

14.   Line 359-360: The low pKa of these NSAIDs,  when present in the acidic inflammatory microenvironment, causes ion trapping of these 361 drugs as they are shifted into the intracellular space and cell membranes [37]. – is this corrective?

15.   Line 375: how is suppression of PGE2 in gastric mucosa supposed to affect intestinal blood flow?  The paper referred to is a review and not a primary source. This does not seem clear.

16.   Line 382: Meaning of this statement is not clear:  The uncoupling of mitochondrial oxidative phosphorylation paired to decrease in mucosal blood flow leads to increased mucosal permeability, and exposure of enterocytes to caustic gastrointestinal contents such as bile and gastric acid.

17.   Line 428: report of 111% bioavailability. This is not really physiologic sense and likely just reflects normally experimental variability. Better just to say bioavailability was complete, or something like that.

18.   Line 438 – this should be related to the long half-life of the drug and it should be explained what this means.

19.   Line 533 – I don’t think carprofen is licensed for horses in Canada. I checked the drug products database and it was not there.

20.   Including drugs and other produced with essentially no data in horses is not necessary or useful.

21.   Dipyrone is not really an atypical NSAID.  It is really a prodrug as they mention.  I would just leave out the COX-3 story as it never really made sense to those working in the area.

22.   Throughout, and in the section at lines 1132- 1139, the value of in vitro COX inhibition numbers is overstated.

23.   Conclusions on uptake – I wonder if there is any data on the reasons for the apparently slow uptake. I don’t think any work has been done on this. 

Reviewer 3 Report

The Clinical Pharmacology and Therapeutic Evaluation of Non-Steroidal Anti-Inflammatory Drugs in Adult Horses

This manuscript provides a narrative review of the pharmacokinetics, efficacy and adverse effect risks of NSAIDs and selected other drug in horses. It appears to be well-referenced and the content is fine. The manuscript would be more useful and readable if the text were interrupted with some tables.

A table describing comparisons of therapeutic effects would be especially welcome and useful. With the current presentation, it is impossible to keep track of how all the drugs compare in efficacy studies, so having a summary table with the various studies and perhaps columns presenting the drugs tested, dosages, disease and model (eg lameness, natural or induced), outcomes measured and results. This would greatly enhance the manuscript and increase its value.

Otherwise it is good. More specific comments:

Line 26-28 This definition of Pk is referenced but maybe not optimal. PK quantifies movement of drugs through the body, but not always the underlying physiologic processes.

Line 28 delete ‘meanwhile”

Line 39 delete “finally”

Line 47 Early retirement from what?

Line 49-50 Recommend deleting the first half of this sentence and just starting with “multiple factors can lead to…”

Line 52 “may be” or are?

Line 56-57 Please be more specific than “lack of”. Are there none, or a few (Legend? Adequan?). it’s not clear if “lack of” means “absence of”

Line 59 symptom or clinical sign?

Line 85: suggest “… as well as reduction of the production and propagation…”

Line 94 maybe add “described below” after SIRS, to indicate that the next paragraph is not a whole change of subject. Might make it easier to follow

Line 109 “it” or “they”?

Line 129 suggest “”…followed by secondary and tertiary febrile responses.”

Lines 137-138 Might be good to add a sentence explain how sequestration in vascular spaces agrees with leukopenias; intuitively it seems like having more cells in the vascular space would increa the number in a blood sample, not decrease it.

Line 140 which contributes to?

Line 147 “…the most widely used analgesic agents by US equine practitioners…”

Line 153 deleted “therefore”

Line 169 pyrazolidine drug class

Line 175-176 last phrase needs a verb

Line 183 bioavailability of

Line 196 adverse effects may be a preferable term to side effects

Line 249 Really appreciate this discussion of the old ideas about different agents for somatic vs visceral pain

Lines 266-299 This is a great discussion.

Line 337 “FDA-approved” is more appropriate than “labelled” in formal writing

Line 352-356 I wonder if this disconnect between plasma and local concentrations should be mentioned earlier, when the PK of NSAIDS is first discussed?

Lines 380-386 This is hard to follow. It’s not clear how a reduction in gastric blood flow would result in increased exposure of enterocytes to caustic GI contents, when the decreased blood flow is in the stomach but the enterocytes are in the intestine. Plus NSAID use risks gastric ulceration, not so much intestinal, right?

Line 416 “…from their structure; they have been developed…”

Line 417 “…they are size -limited…”

Line 475 “Has” not “have”, safety is singular.

Line 499 Approved not labelled, and in which countries? 5 are listed

Line 498 “reduced” implies that it used to be higher. Perhaps just say low or minimal.

Line 533 Does Canada use the term licensed instead of approved?

Line 560 “its” not “it’s” “It’s” means only it is or it has, it is not possessive.

Line 575-576 2/6 is not greater than 3/6 or 4/6

Line 637 Only one approved CBD product for horses, or overall?

Line 638 “Therefore” implies causality, would suggest However instead

Line 656 “data are” data is plural

Line 661 “…compared to other NSAIDs” worth mentioning that plenty of folks do not consider dipyrone anti-inflammatory enough to be an NSAID

Line 700 not sure ‘lesions’ is the word here, perhaps ’conditions ultimately requiring surgery’, or something like that

Line 746-7 “similarly”

Line 766 please name the other treatments

Line 768-769 please describe the objective lameness parameter

Line 796 and elsewhere: “utilize” is not a synonym for “use”. “Utilize” means to use in a way other than for its intended use- like you might utilize a paper clip to pick a lock. If you are using the paper clip to hold papers together, you are using it. I believe, here and elsewhere, “use” is more appropriate.

Line 821 might say “treatment regimen” to improve clarity about what you are referencing

Lines 834 -837 you names 3 teats and 2 types of pain so it is not clear what “respectively” means inthis context”

Line 874 and 879 Is the “most popular” model only used in one study?

Line 881 and 884 not clear what “these models” refers to

Line 892-893 Acetaminophen in not an NSAID

Line 907 “This model has also been used to…”

Line 909 Might want to explain what mean difference in head height means

Line 925 Which objective lameness evaluation?

Line 927 “when circling” is probably more clear than “on the lunge”

 Lie 856 “described above”

Line 957 “response of systemic inflammation to drug treatment”

It’s not entirely clear how section 7 differs from section 6? The methods n 7 are also used for research

Line 1000 You described challenges related to agreement, not to quantifying varying degrees of severity of lameness

Sections 7.2.1 and 7.2.2 It would be more clear if you first described BMIS before describing how it compares to force plates.

Line 1052 I think you mean quantitative, not quantifiable

Line 1078 veterinary evaluation or evaluation by veterinarians

Lines 1103-1105 It is surprising to read that HF is directly correlated to adrenalin then to read that HF is also known as Vagal tone

Lines 1187-1180 Many people do not consider acetaminophen and dipyrone to be NSAIDs

Round 2

Reviewer 1 Report

The authors have done a very nice job with this revision and have addressed my comments and suggestions to my satisfaction. I have no additional comments or suggestions.

Author Response

We would like to thank you for your continued time and efforts reviewing our manuscript. 

Reviewer 2 Report

The authors were very open to most of the suggestions previously made and the manuscript is markedly improved as a result. The authors have brought a lot of information together in one place.

While there are a few things I might still quibble with, most of these are matters of interpretation or personal preference and should not inhibit publication.

Since they shifted from "atypical NSAIDs" to "non-conventional" NSAIDs in the text, they might want to use the same language in Figure 1.

Author Response

We would like to thank you for your continued time and efforts reviewing our manuscript. We have updated figure 1 to replace the term “atypical NSAIDs” with “non-conventional NSAIDs” per your recommendation.

Reviewer 3 Report

Thank you to the authors for their responses to my previous comments.

I do not believe that the major additions to the manuscript enhance it. The new section Pain, Nociception, and the cyclo-oxygenase pathway is too lengthy, too detailed, and a little off topic. A briefer discussion with more focus on the effects of NSAIDs would be more appropriate and relevant, and more useful to the average reader. You can always refer the reader elsewhere for publications with greater detail about the pathophysiology of pain. I apologize if this was requested by another reviewer- unlike with some other journals, I am not able to see other reviews.

 The PK tables are fine, but still not as useful as tables describing efficacy, particularly since efficacy is a focus of the manuscript text. Despite the different outcome measures this can be done- just include a column listing the outcomes measured for each study- with footnotes if greater description needed- then a column or columns listing which outcomes were affected and which were not. This would be of greater utility to the average reader than a PK table. I have seen this done in review articles plenty of times, and it really is helpful and worthwhile. It could be just a single table for all of the drugs discussed, or separate tables.

Line 225: Symptoms or clinical signs?

Line 254: Recommend staying away from “tend to” in scientific writing, it’s vague. Has the effect been demonstrated, or is it believed that the drugs behave this way?

Lines 269-280: This discussion should also include the role of measuring PGE and TXB in the determination of COX-1 and COX-2 effects, since you discuss PGE and TXB inhibition and ratios later in the manuscript in the descriptions of individual drugs.

Line 346: might be more clear to say “a carboxylic acid drug and…” as is it almost sounds like flunixin inhibits carboxylic acid and COX.

Line 355: Similarly

Line 886: Again, might be useful to explain the meaning/implication of head height in lame horses, even parenthetically

Sections 6 and 7: Again, it would make more sense to describe measures of pain before using them to describe drug effects instead of after.

Author Response

We would like to thank you for your continued time and efforts reviewing our manuscript. We have addressed some of your concerns and revision requests as comments below.

Comment: “I do not believe that the major additions to the manuscript enhance it. The new section Pain, Nociception, and the cyclo-oxygenase pathway is too lengthy, too detailed, and a little off topic. A briefer discussion with more focus on the effects of NSAIDs would be more appropriate and relevant, and more useful to the average reader. You can always refer the reader elsewhere for publications with greater detail about the pathophysiology of pain. I apologize if this was requested by another reviewer- unlike with some other journals, I am not able to see other reviews.”

Response: There appears to be a conflict between the reviewers regarding the major additions to the introduction. While these sections (expansion on pain physiology and the COX pathway) were added to the introduction to address comments from reviewers 1 & 2, we understand your view that the additions have made the introduction too long and overly detailed. While we agree with you that the reader could find this information elsewhere, we propose to leave the introduction as is to provide the reader with the appropriate context to understand the rest of the review without having to consult other sources and to align with comments from the other reviewers.

Comment: “The PK tables are fine, but still not as useful as tables describing efficacy, particularly since efficacy is a focus of the manuscript text. Despite the different outcome measures this can be done- just include a column listing the outcomes measured for each study- with footnotes if greater description needed- then a column or columns listing which outcomes were affected and which were not. This would be of greater utility to the average reader than a PK table. I have seen this done in review articles plenty of times, and it really is helpful and worthwhile. It could be just a single table for all of the drugs discussed, or separate tables.”

Response: After consultation with the co-authors and the section editor for the manuscript we have not made this requested change as we feel this is out of scope of the current manuscript and would be better suited to a systematic meta-analysis of NSAID efficacy in horses.

Comment: Line 225: Symptoms or clinical signs?

Response: We have made this change (line 224)

Line 254: Recommend staying away from “tend to” in scientific writing, it’s vague. Has the effect been demonstrated, or is it believed that the drugs behave this way?

Response: We have changed this sentence to be more definitive (line 611)

Lines 269-280: This discussion should also include the role of measuring PGE and TXB in the determination of COX-1 and COX-2 effects, since you discuss PGE and TXB inhibition and ratios later in the manuscript in the descriptions of individual drugs.

Response: We have included the reference back to TXB2 and PGE2 in this section (line 623-624). Additionally, by moving sections 6 and 7 as you have recommended, the reader is better prepared to understand these concepts when they encounter them in the individual drug sections.

Line 346: might be more clear to say “a carboxylic acid drug and…” as is it almost sounds like flunixin inhibits carboxylic acid and COX.

Response: We have restructured this sentence to improve its clarity (line 697).

Line 355: Similarly

Response: We have made this change (line 712)

Line 886: Again, might be useful to explain the meaning/implication of head height in lame horses, even parenthetically

Response: We have made this change (line 1223).

Sections 6 and 7: Again, it would make more sense to describe measures of pain before using them to describe drug effects instead of after.

Response: We agree that this would improve the manuscript, and we have made this change.